# Sculpting Subspaces: Constrained Full Fine-Tuning in LLMs for Continual Learning

**Nikhil Shivakumar Nayak**[1,5][*]**, Krishnateja Killamsetty**[2]**, Ligong Han**[1,5]**,**
**Abhishek Bhandwaldar**[2,5]**, Prateek Chanda**[3]**, Kai Xu**[1,5]**, Oleg Silkin**[1]**,**
**Mustafa Eyceoz**[1]**, Hao Wang**[1,5]**, Aldo Pareja**[1,5]**, Akash Srivastava**[4,5]

[1]Red Hat AI Innovation   [2]IBM Research   [3]IIT Bombay   [4]Core AI, IBM
[5]MIT-IBM Watson AI Lab

## Abstract

Continual learning in large language models (LLMs) is prone to catastrophic forgetting, where adapting to new tasks significantly degrades performance on previously learned ones. Existing parameter-efficient methods often limit model expressivity or introduce new parameters per task, creating scalability issues. To address these limitations, we introduce **Orthogonal Subspace Fine-Tuning (OSFT)**, a novel parameter-efficient approach for continual learning. OSFT leverages adaptive singular value decomposition (SVD) to dynamically identify and preserve critical, high-rank parameter subspaces that encode prior knowledge. All updates for new tasks are constrained to be strictly orthogonal to these preserved subspaces, which minimizes interference while maintaining a fixed parameter count and avoiding the need to store task-specific gradients. We extensively evaluate OSFT on standard continual learning benchmarks using both encoder-decoder (T5-Large) and decoder-only (LLaMA-2 7B, Mistral-7B) models across diverse tasks. Empirically, our method achieves a state-of-the-art trade-off between learnability and knowledge retention, dominating the Pareto frontier, with **up to 7% higher** average accuracy than recent baselines like O-LoRA, and **reduces forgetting to near-negligible levels**. It notably maintains the model's general linguistic capabilities, instruction-following, and safety throughout the learning process. OSFT provides a practical, theoretically grounded, and scalable solution that effectively balances model plasticity and knowledge retention for continual learning in LLMs. Code is available at `https://github.com/Red-Hat-AI-Innovation-Team/mini_trainer`.

## 1 Introduction

Large language models (LLMs), such as GPT-3 (Brown et al., 2020), PaLM (Chowdhery et al., 2023), and LLaMA-2 (Touvron et al., 2023), have achieved remarkable successes across a broad range of natural language tasks. However, deploying these models in dynamic, real-world scenarios presents a fundamental challenge: how can we efficiently adapt them to new tasks and evolving data distributions without losing their valuable pre-trained knowledge?

Consider an enterprise LLM that must continuously learn new product information, regulatory updates, and domain-specific terminology. Traditional full fine-tuning—updating all billions of parameters—not only incurs prohibitive computational costs but also leads to *catastrophic forgetting* (McCloskey & Cohen, 1989; Kirkpatrick et al., 2017), where the model's performance on previously learned tasks deteriorates dramatically. This creates an impossible choice: maintain separate models for each task (multiplying infrastructure costs) or accept degraded performance on earlier capabilities.

**Parameter-Efficient Fine-Tuning (PEFT)** methods like Adapters (Houlsby et al., 2019) and LoRA (Hu et al., 2022) reduce computational costs by freezing the pre-trained model and introducing small

---

[*]Correspondence to: Nikhil Shivakumar Nayak `<nayak.nikhil2608@gmail.com>`.

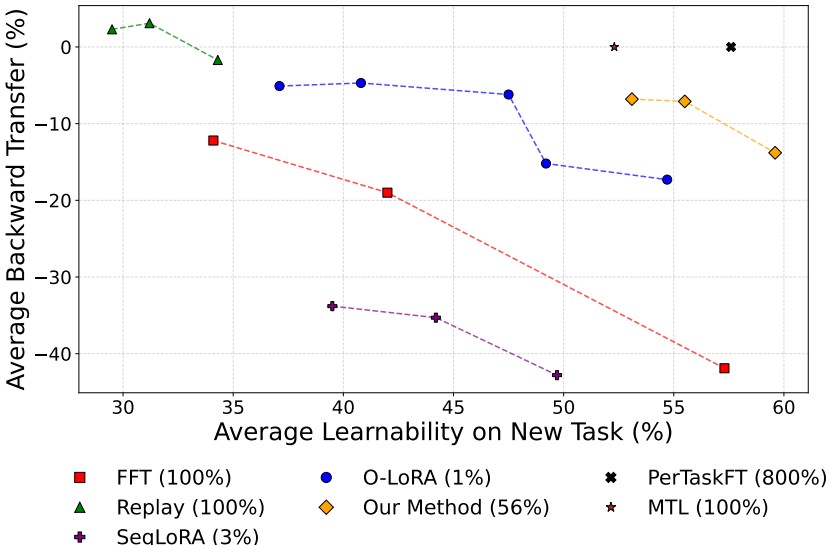

Figure 1: **Pareto frontier on TRACE Benchmark.** Each curve shows the trade-off between average immediate task accuracy and average backward transfer (BT, our forgetting metric; more negative BT means more forgetting) for a given method; **higher values indicate better performance on both axes**. Our approach dominates the frontier achieving the best overall performance (*learnability – forgetting*) while being parameter efficient. Legend entries report the average fraction of trainable parameters used (approximate). For this figure, all methods are run on LLaMA-2-7B-Chat with the same training schedule; we sweep a small grid over key hyper-parameters per method: learning rate $\in \{10^{-3}, 10^{-4}, 10^{-5}\}$ for FFT, replay buffer size $\in \{5\%, 10\%, 15\%\}$ of previous-task data for Replay, LoRA rank $\approx \{2\%, 3\%, 4\%\}$ of the matrix dimension for SeqLoRA, LoRA rank $\approx \{0.5\%, 1.0\%, 1.5\%\}$ for O-LoRA (all with orthogonality regularization $\lambda = 0.5$) and two additional O-LoRA points at fixed rank $1.0\%$ with $\lambda \in \{0.2, 1.0\}$, and average effective trainable rank $\approx \{50\%, 56\%, 62\%\}$ across the 8 tasks for **Ours**. *PerTaskFT* and *MTL* serve as upper bounds, obtained by training a separate model per task and by joint multi-task training on the full data, respectively.

trainable modules. However, their restricted parameter budget limits adaptation capacity, and they struggle in continual learning—either accumulating new modules per task or requiring complex merging strategies. **Continual Learning approaches** like EWC (Kirkpatrick et al., 2017) and O-LoRA (Wang et al., 2023a) attempt to address forgetting but either provide only soft constraints that slow rather than prevent forgetting, or operate within fixed low-rank subspaces that may not align with the model's natural capacity distribution.

The key insight missing from these approaches is that *not all parameter directions are created equal*. Recent work (Sharma et al., 2023) reveals that neural network weight matrices contain substantial redundancy—many parameter directions, particularly those with small singular values, contribute minimally to model behavior. This suggests we could identify and repurpose these "dormant" directions for new tasks while preserving critical knowledge-encoding directions.

Building on this insight, we propose **Orthogonal Subspace Fine-Tuning**, a novel parameter-efficient method that fundamentally rethinks how models adapt to new tasks. As illustrated in Figure 2, our approach operates through three synergistic mechanisms:

1. **Adaptive Subspace Identification**: We decompose each layer via SVD to separate critical knowledge-bearing directions (high singular values) from underutilized capacity (low singular values) that can be safely repurposed.

2. **Importance-Guided Allocation**: We measure each layer's importance via input-output cosine similarity. High-similarity layers that primarily preserve features (e.g., early attention layers) receive more protected singular directions to maintain stability. Low-similarity layers that transform representations (e.g., final MLPs) are allocated more adaptable ca-

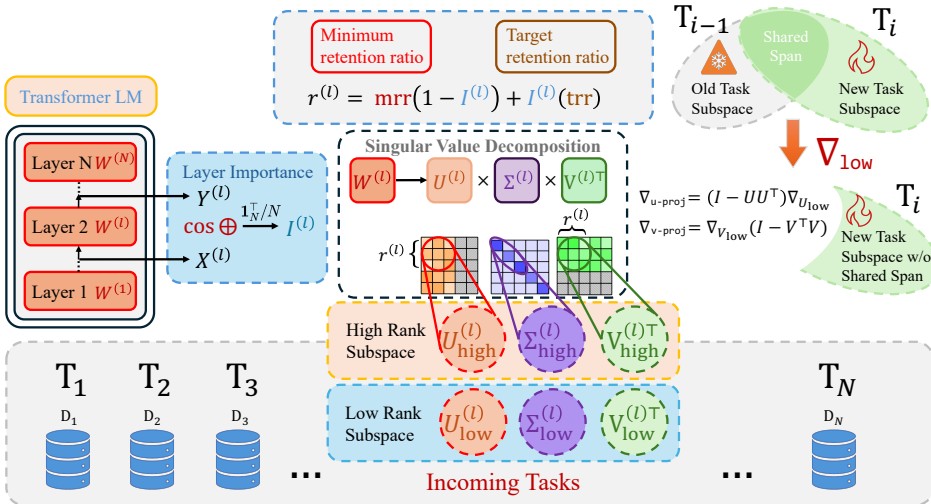

Figure 2: **Overview of our Adaptive SVD-based Continual Fine-tuning Method.** For each parameter matrix in the network, we perform SVD decomposition to identify high-rank components (associated with larger singular values) that encode crucial knowledge from previous tasks, and low-rank components (associated with smaller singular values) that contribute minimally to model performance. When learning a new task, gradient updates are projected onto the low-rank subspace orthogonal to previous task representations, allowing full parameter updates while minimizing catastrophic forgetting.

pacity for learning new tasks. This adaptive allocation automatically balances stability and plasticity across the network according to each layer's functional role.

3. **Orthogonal Gradient Projection**: We constrain all gradient updates to remain strictly orthogonal to preserved subspaces, creating an impenetrable barrier against forgetting.

Unlike existing methods that either waste parameters (full fine-tuning), sacrifice expressivity (fixed adapters), or accumulate modules (progressive approaches), our design achieves all desired properties simultaneously. Figure 1 empirically validates this claim—our method dominates the Pareto frontier, achieving superior learning with minimal forgetting while minimizing the parameter usage.

## 1.1 OUR CONTRIBUTIONS

**1. An orthogonal subspace approach to parameter-efficient fine-tuning:** We propose a theoretically grounded method that partitions weight matrices via adaptive SVD to identify and reuse low-importance parameter subspaces with minimal interference. This effectively balances the plasticity needed for new tasks with stability to retain prior knowledge.

**2. Adaptive capacity allocation without extra memory:** Our method dynamically allocates parameter budgets across layers based on their functional role while maintaining a fixed footprint, avoiding new modules or stored gradients for each task and thus scaling gracefully to many tasks.

**3. State-of-the-art performance on diverse tasks:** We demonstrate consistent gains across classification, generation, math, and reasoning benchmarks using T5-Large, LLaMA-2 7B, and Mistral-7B models. Our approach achieves better accuracy, stronger knowledge retention, and nearly negligible forgetting—while preserving general linguistic capabilities, instruction-following, and safety.

**4. Thorough empirical and theoretical validation:** We provide in-depth analyses verifying the effective repurposability of low-rank subspaces, showing that these directions can be used for new tasks without degrading old ones. Our experiments confirm practical robustness while theoretical analysis proves tighter bounds on catastrophic forgetting.

The remainder of this paper is structured as follows. Section 2 reviews relevant literature. Section 3 presents our algorithm in detail. Section 4 provides experimental validation. Section 5 concludes with key insights and future directions.

## 2 RELATED WORK

Continual learning in large language models aims to acquire new knowledge without catastrophically forgetting the old. Existing methods typically achieve this by either restricting *which* parameters are updated or by constraining *how* full-parameter updates are performed.

**Parameter-restricted approaches** isolate task knowledge by modifying only a small subset of weights. Parameter-Efficient Fine-Tuning (PEFT) methods like Adapters (Houlsby et al., 2019) and LoRA (Wang et al., 2023a; Liang & Li, 2024) freeze the base model and train a few new parameters per task. While this isolates updates, it can limit expressiveness and adds parameter overhead that scales with the number of tasks. Recent SVD-based variants such as MiLoRA (Wang et al., 2025) and PiSSA (Meng et al., 2024) further factor weight matrices and interpret low- or high-singular-value components as LoRA-style adapters: MiLoRA freezes the high-singular-value components and updates only the low-singular-value components, while PiSSA freezes the low-singular-value components and updates the high-singular-value components instead. OSFT is structurally closest to MiLoRA in that it also fine-tunes only low-singular-value components, but differs in two key ways: (i) we constrain updates via an orthogonal subspace projection so that gradients remain in the complement of the preserved subspace, and (ii) we select ranks adaptively on a per-layer basis using an input–output cosine similarity importance score rather than fixing a global rank for all layers. Similarly, **sparse fine-tuning** methods (Panda et al., 2024; Bhardwaj et al., 2025) update a small fraction of the original weights, but their selection often relies on heuristics like gradient magnitude. Our approach differs by using Singular Value Decomposition (SVD) to identify critical subspaces, a choice theoretically grounded in the connection between singular values and the loss landscape's curvature, offering a more principled selection method.

**Update-constraining approaches** modify all parameters but impose constraints to protect prior knowledge. **Regularization** methods such as EWC (Kirkpatrick et al., 2017) penalize changes to important weights but cannot fully prevent interference, leading to gradual performance decay. A closer line of work, **gradient projection**, constrains updates to be orthogonal to subspaces learned from past tasks. However, leading methods like GPM (Saha et al., 2021) and SGP (Saha & Roy, 2023) derive these subspaces from task *activations*. This creates a critical bottleneck, as their memory requirements grow linearly with the number of tasks, rendering them impractical for billion-parameter models. Other full-parameter strategies like standard fine-tuning (Luo et al., 2025) and model merging (Jang et al., 2024; Yadav et al., 2023) represent extremes of catastrophic forgetting or prohibitive computational cost, respectively.

OSFT's design is motivated by the limitations of the following existing strategies.

**Fixed-Rank and Regularization Methods.** A simpler approach might be a fixed-rank projection (e.g., freezing the **top k** singular vectors in all layers), but this ignores layer heterogeneity and can either over-preserve (hurting plasticity) or under-preserve (causing forgetting). Regularization methods like EWC are often insufficient for LLMs, as their diagonal Hessian approximations fail to capture the complex, non-diagonal curvature of the loss landscape, leading to subpar knowledge retention (Ritter et al., 2018; Heckel, 2022; Kruengkrai & Yamagishi, 2022).

**Activation-Based Projection (GPM/SGP).** A related line of work uses gradient projection but differs fundamentally. Methods like Gradient Projection Memory (GPM) (Saha et al., 2021) and Scaled Gradient Projection (SGP) (Saha & Roy, 2023) perform SVD on task *activations* to build a basis for important subspaces. Our approach differs in four key ways:

1. **Object of SVD:** We perform SVD directly on model **weights** to approximate high-curvature directions, whereas GPM/SGP operate on **activations**.

2. **Memory Scalability:** Our memory overhead is **constant**, as we only store the singular vectors of the current weights. In contrast, GPM/SGP accumulate activation-derived bases, causing memory to grow linearly with the number of tasks. This makes them impractical for billion-parameter LLMs.

3. **Adaptive Ranks:** Our method uses a layer-wise importance score to adaptively allocate rank, offering a more flexible balance of stability and plasticity.

4. **Target Scale:** OSFT is the first weight-SVD projection method validated on billion-parameter LLMs, whereas prior methods were demonstrated on smaller-scale models.

Our work introduces a method that combines the expressive capacity of full-model updates with a scalable and theoretically grounded constraint mechanism. Unlike methods that limit updates to small parameter subsets, we leverage the entire model, and unlike prior projection methods, our weight-based SVD approach maintains constant memory overhead, making it practical for continual learning in state-of-the-art language models.

## 3 METHODOLOGY

Our approach, Orthogonal Subspace Fine-Tuning (OSFT), addresses continual learning in large language models by leveraging adaptive low-rank updates guided by Singular Value Decomposition (SVD). We strategically preserve critical knowledge from previous tasks by constraining parameter updates away from dominant (high-rank) singular directions, while enabling model adaptation within complementary (low-rank) directions.

### 3.1 PROBLEM SETUP AND NOTATION

Let the parameters of an LLM be denoted as:

$$\theta = \{\mathbf{W}^{(1)}, \mathbf{W}^{(2)}, \ldots, \mathbf{W}^{(L)}\},$$

where each $\mathbf{W}^{(l)} \in \mathbb{R}^{d_O^{(l)} \times d_I^{(l)}}$ represents the weight matrix of layer $l$. Practical deployments involve matrices with millions or billions of parameters, underscoring the necessity of efficient continual updates.

Given sequential tasks $\{\mathcal{D}_1, \mathcal{D}_2, \ldots, \mathcal{D}_T\}$, each defined by data pairs $\{(x_i^t, y_i^t)\}_{i=1}^{n_t}$, our goal is to sequentially adapt parameters $\theta$ to task $\mathcal{D}_t$ without significant performance degradation on previously learned tasks $\mathcal{D}_1, \ldots, \mathcal{D}_{t-1}$. Training repeatedly from scratch is computationally prohibitive, necessitating efficient incremental updates.

### 3.2 LOW-RANK AND HIGH-RANK SUBSPACES VIA SVD

Extensive empirical evidence shows neural network parameters possess substantial redundancy (Sharma et al., 2023; Hartford et al., 2024), where directions associated with small singular values minimally impact critical model knowledge. Conversely, larger singular values typically encapsulate vital knowledge. Empirical verification of this low-rank assumption is in Appendix A.4. Leveraging this observation, we propose:

> *Projecting parameter updates away from high singular-value directions, preserving previously acquired knowledge, and utilizing low singular-value directions for adaptation to new tasks.*

Formally, we perform Singular Value Decomposition (SVD) on each weight matrix $\mathbf{W}^{(l)}$ at layer $l$:

$$\mathbf{W}^{(l)} = \mathbf{U}^{(l)} \Sigma^{(l)} (\mathbf{V}^{(l)})^\top, \tag{1}$$

where singular values in $\Sigma^{(l)}$ are sorted in descending order. We compute this decomposition once per task, adding minimal overhead compared to full model training.

### 3.3 DETERMINING LAYER IMPORTANCE VIA INPUT–OUTPUT SIMILARITY

Inspired by AdaSVD (Li et al., 2025), we quantify layer importance using cosine similarity between a layer's input activations $\mathbf{X}^{(l)}$ and its linear outputs $\mathbf{Y}^{(l)} = \mathbf{W}^{(l)} \mathbf{X}^{(l)}$. Specifically, when evaluating layer importance for task $t + 1$, we compute the similarity using data samples from the previous

task $t$ as follows:

$$I^{(l)} = \frac{1}{N} \sum_{i=1}^{N} \text{cosine\_similarity}(\mathbf{X}_i^{(l)}, \mathbf{Y}_i^{(l)}) \tag{2}$$

where $N$ denotes the number of data samples from task $t$. Higher similarity indicates minimal directional change, signifying that the layer predominantly preserves rather than transforms activation representations. Such layers are essential for retaining features and ensuring stable propagation of information across tasks. Importance scores are also normalized to have an average of one across layers: $\frac{1}{L} \sum_{l=1}^{L} I^{(l)} = 1$. While empirically we observe $I^{(l)}$ is consistently positive, for robustness, we clip any negative raw cosine similarity values to zero before normalization.

We explored alternative rank-approximation strategies, including LASER (Sharma et al., 2023), SPECTRUM's Marchenko–Pastur thresholding (Hartford et al., 2024), and entropy-based effective rank (Roy & Vetterli, 2007). In a continual learning setting, these approaches either fail to capture layer-wise variability under sequential tasks or do not yield stable thresholds across heterogeneous datasets. An alternative notion of representational similarity is centered kernel alignment (CKA) (Kornblith et al., 2019), which compares activations across models or layers. CKA emphasizes a different aspect of representation geometry focusing on cross-representation similarity, while cosine similarity is designed to capture how tightly a layer's outputs are aligned with its inputs. A CKA-based variant of our importance measure is a natural extension.

### 3.4 ADAPTIVE RANK SELECTION

Given the importance of the layer $I^{(l)}$, we introduce two hyperparameters controlling the retention of singular vectors:

- **Minimum Retention Ratio (mrr)**, ensuring minimal essential retention even for the least critical layers.
- **Target Retention Ratio (trr)**, defining the upper retention bound for highly critical layers.

The fraction of singular vectors preserved at each layer is computed as:

$$r_{\text{frac}}^{(l)} = \text{mrr} + I^{(l)}(\text{trr} - \text{mrr}). \tag{3}$$

The number of singular vectors to retain is $k^{(l)} = \lfloor r_{\text{frac}}^{(l)} \cdot \min(d_O^{(l)}, d_I^{(l)}) \rfloor$. The singular vectors are then partitioned into high-rank ($\mathbf{U}_{\text{high}}^{(l)}, \mathbf{V}_{\text{high}}^{(l)}$) and low-rank subspaces. In practice, we found that setting $\text{mrr} = 0.1$ and $\text{trr} = 0.8$ yields robust performance across our benchmarks. Our method is not overly sensitive to these values; ablation studies show that while performance degrades significantly if retention is too aggressive, mild perturbations ($\pm 0.05$) result in minimal ($<1\%$) accuracy changes. For new applications, we recommend starting with these defaults and performing a small grid search on the first task. See Appendix A.9 for the complete ablation study.

### 3.5 ORTHOGONAL GRADIENT UPDATES IN LOW-RANK SUBSPACE

To minimize catastrophic forgetting, we enforce updates to lie within the low-rank subspace orthogonal to the high-rank directions. This is achieved by projecting the gradients:

$$\nabla \mathbf{W}_{\text{proj}}^{(l)} = \nabla \mathbf{W}^{(l)} - \mathbf{U}_{\text{high}}^{(l)} \left( (\mathbf{U}_{\text{high}}^{(l)})^\top \nabla \mathbf{W}^{(l)} \mathbf{V}_{\text{high}}^{(l)} \right) (\mathbf{V}_{\text{high}}^{(l)})^\top. \tag{4}$$

Here, $\mathbf{U}_{\text{high}}^{(l)} \in \mathbb{R}^{d_O \times k^{(l)}}$ and $\mathbf{V}_{\text{high}}^{(l)} \in \mathbb{R}^{d_I \times k^{(l)}}$ are the dense matrices containing the top $k^{(l)}$ singular vectors. This operation computes the component of the gradient that lies within the high-rank subspace and subtracts it, ensuring the final update is strictly orthogonal to the preserved directions.

### 3.6 ORTHOGONAL UPDATES VIA REPARAMETERIZATION AND GRADIENT HOOKS

Our goal is to confine all updates to the low-rank subspace, making them orthogonal to the frozen high-rank directions. While one can achieve this by projecting the full weight gradient, a more computationally efficient and elegant solution is implemented by reparameterizing the weights and using gradient hooks. This process involves two main steps:

1. **Reparameterization and Freezing:** After performing SVD on a weight matrix $\mathbf{W}^{(l)}$, we replace it with its underlying SVD components.

   - The high-rank components $(\mathbf{U}_{\text{high}}^{(l)}, \Sigma_{\text{high}}^{(l)}, \mathbf{V}_{\text{high}}^{(l)})$ are registered as frozen buffers in the model (i.e., non-trainable).
   - The low-rank components $(\mathbf{U}_{\text{low}}^{(l)}, \Sigma_{\text{low}}^{(l)}, \mathbf{V}_{\text{low}}^{(l)})$ are registered as new trainable parameters.

   During the forward pass, the full weight is reconstructed on-the-fly ($\mathbf{W} = \mathbf{W}_{\text{high}} + \mathbf{W}_{\text{low}}$). During backpropagation, gradients are only computed for the trainable low-rank SVD components.

2. **Maintaining Orthogonality with Gradient Hooks:** Simply training the low-rank components could cause their basis vectors (the columns of $\mathbf{U}_{\text{low}}$ and $\mathbf{V}_{\text{low}}$) to "drift" and lose their perfect orthogonality with the frozen high-rank basis vectors. To prevent this, we attach a gradient hook to the trainable parameters. After the gradients (e.g., $\nabla \mathbf{U}_{\text{low}}^{(l)}$) are computed, this hook projects them to be orthogonal to the high-rank basis vectors (e.g., $\mathbf{U}_{\text{high}}^{(l)}$). This acts as a maintenance step, guaranteeing the mathematical integrity of the subspaces throughout training.

Our OSFT procedure is summarized in Algorithm 1.

## 3.7 COMPUTATIONAL AND MEMORY ANALYSIS

Our method is designed to be computationally efficient and scalable. The primary additional cost is the SVD, which is performed once per layer per task before training begins. Refer to Appendix A.11 for detailed computational cost and memory efficiency analysis.

## 3.8 THEORETICAL JUSTIFICATION OF ADAPTIVE RANK SELECTION

We rigorously justify our adaptive rank selection method through a formal theoretical analysis using a second-order Taylor expansion of the task-specific loss landscape, detailed in Appendix A.3. This analysis explicitly demonstrates that preserving parameter directions associated with the highest Hessian eigenvalues—representing directions of greatest curvature—effectively minimizes catastrophic forgetting.

However, explicitly computing and decomposing the Hessian is computationally prohibitive for large-scale language models. Therefore, we employ an efficient approximation inspired by empirical evidence from Haink (2023), who show a robust correlation between the Hessian's largest eigenvalues and the largest singular values of the model's weight matrices. By retaining the top singular vectors—corresponding to critical knowledge from previous tasks—we effectively approximate freezing the high-curvature Hessian directions while allowing updates within the subspace defined by lower singular values.

Further supporting our approach, empirical findings (Sharma et al., 2023; Li et al., 2025) highlight that layers with higher input-output similarity exhibit significantly greater Hessian curvature. Our adaptive layer-wise rank allocation strategically exploits this property: layers identified as crucial (high input-output similarity) receive greater singular vector retention, thereby preserving essential knowledge.

## 4 EXPERIMENTAL RESULTS

We comprehensively evaluate our adaptive SVD-based continual learning method on established continual learning benchmarks, comparing it extensively with recent state-of-the-art (SOTA) baselines, notably O-LoRA Wang et al. (2023a). Our experiments aim to demonstrate the effectiveness, scalability, and practicality of our approach in realistic continual learning scenarios. For all benchmarks, we use fixed task sequences from prior work and report results averaged across multiple orders. To provide deeper insight into learning and forgetting, we include per-task accuracies (both immediately after training and at the end of the sequence) in Appendix A.12. We compare against

recent SVD baselines MiLoRA and PiSSA; results are reported in Appendix A.6. We also compare against the SOTA sparse fine-tuning method LoTA (Panda et al., 2024), which uses sparsity masks to preserve task-specific information; results and analysis are provided in Appendix A.7.

## 4.1 BENCHMARKS AND EVALUATION PROTOCOL

We adopt two widely-used benchmarks reflecting varying levels of complexity and task diversity:

**Standard Continual Learning Benchmark (5 Tasks)** introduced by Zhang et al. (2015), consisting of classification tasks: AG News, Amazon Reviews, Yelp Reviews, DBpedia, and Yahoo Answers.

**Extended Continual Learning Benchmark (15 Tasks)**, introduced by Razdaibiedina et al. (2023), combining tasks from multiple sources, including GLUE (Wang et al., 2019) (MNLI, QQP, RTE, SST-2), SuperGLUE (Wang et al., 2020) (WiC, CB, COPA, MultiRC, BoolQ), and IMDB, along with the original 5-task benchmark.

**TRACE Benchmark (8 Tasks).** In addition, we evaluate on TRACE (Wang et al., 2023b), an 8-task instruction-tuning continual learning benchmark covering domain-specific tasks, multilingual capabilities, code generation, and mathematical reasoning. Following TRACE, we report two metrics: *Average Accuracy (AA)* and *Average Backward Transfer (BT)*. Let $A_{i,j}$ denote the accuracy on task $j$ after training on task $i$ in a sequence of $T$ tasks. We define:

$$\text{AA} = \frac{1}{T} \sum_{j=1}^{T} A_{T,j}, \quad \text{BT} = \frac{1}{T-1} \sum_{j=1}^{T-1} \left( A_{T,j} - A_{j,j} \right). \tag{5}$$

We interpret *forgetting* as $-\text{BT}$: more negative BT corresponds to more forgetting, while values closer to zero indicate better retention of earlier tasks.

We evaluate two popular large language model architectures, T5-Large (encoder-decoder) and LLaMA-2 7B (decoder-only), using the widely-adopted metric of Average Accuracy (AA), computed across all tasks after training on the final task. To ensure robustness, we follow standard protocols, averaging results over three independent runs with randomly permuted task sequences. Implementation details, hardware configurations, and training hyperparameters for both T5-Large and LLaMA-2 7B models are provided in Appendix A.10.

## 4.2 BASELINE METHODS

We position our method clearly against representative continual learning paradigms:

- **Sequential full-model fine-tuning (SeqFT)**: serves as a lower-bound baseline, prone to catastrophic forgetting.

- **Parameter-efficient LoRA variants** including SeqLoRA, IncLoRA, and the recent SOTA, O-LoRA Wang et al. (2023a), which utilize low-rank adapters.

- **Replay-based approaches**, such as standard replay buffers.

- **Regularization methods**, including Elastic Weight Consolidation (EWC) Kirkpatrick et al. (2017) and Learning without Forgetting (LwF) Li & Hoiem (2017).

- **Prompt-based techniques**, including L2P Wang et al. (2022) and ProgPrompt Razdaibiedina et al. (2023).

- **Model-merging methods**: we include SLERP (Jang et al., 2024) and TIES (Yadav et al., 2023). Both methods operate by combining separate per-task models rather than maintaining a single continually updated model.

- **PerTaskFT**: trains a separate model per task, offering strong performance but requiring extensive computational resources and storage.

- **Multi-task Learning (MTL)**: trains a single model simultaneously on all tasks, representing an ideal upper bound by relaxing continual learning constraints.

Table 1: Comparison of Average Accuracy (%) across standard continual learning benchmarks using the T5-Large model.

| Method | 5-Task CL Benchmark | | | | 15-Task CL Benchmark | | | |
|---|---|---|---|---|---|---|---|---|
| | Order-1 | Order-2 | Order-3 | avg | Order-4 | Order-5 | Order-6 | avg |
| SeqFT | 18.9 | 24.9 | 41.7 | 28.5 | 7.4 | 7.4 | 7.5 | 7.4 |
| SeqLoRA | 44.6 | 32.7 | 53.7 | 43.7 | 2.3 | 0.6 | 1.9 | 1.6 |
| IncLoRA | 66.0 | 64.9 | 68.3 | 66.4 | 63.3 | 58.5 | 61.7 | 61.2 |
| Replay | 55.2 | 56.9 | 61.3 | 57.8 | 55.0 | 54.6 | 53.1 | 54.2 |
| EWC | 48.7 | 47.7 | 54.5 | 50.3 | 45.3 | 44.5 | 45.6 | 45.1 |
| LwF | 54.4 | 53.1 | 49.6 | 52.3 | 50.1 | 43.1 | 47.4 | 46.9 |
| L2P | 60.3 | 61.7 | 61.1 | 60.7 | 57.5 | 53.8 | 56.9 | 56.1 |
| LFPT5 | 67.6 | 72.6 | 77.9 | 72.7 | 70.4 | 68.2 | 69.1 | 69.2 |
| O-LoRA | **75.4** | **75.7** | 76.3 | 75.8 | **72.3** | 64.8 | 71.6 | 69.6 |
| **OSFT (ours)** | 75.3 | 74.0 | **78.4** | **75.9** | 71.6 | **69.6** | **72.7** | **71.3** |
| SLERP | 40.5 | 43.0 | 45.8 | 43.1 | 2.4 | 1.5 | 2.7 | 2.2 |
| TIES | 35.0 | 38.5 | 37.8 | 37.1 | 7.8 | 7.1 | 5.8 | 6.9 |
| ProgPrompt | 75.2 | 75.0 | 75.1 | 75.1 | 78.0 | 77.7 | 77.9 | 77.9 |
| PerTaskFT | 70.0 | 70.0 | 70.0 | 70.0 | 78.1 | 78.1 | 78.1 | 78.1 |
| MTL (Upper Bound) | 80.0 | 80.0 | 80.0 | 80.0 | 76.5 | 76.5 | 76.5 | 76.5 |

Table 2: TRACE benchmark performance using LLaMA-2-7B-Chat. Average Accuracy (AA) and Backward Transfer (BT) percentages are reported.

| Method | AA (%) | BT (%) |
|---|---|---|
| SeqFT | 23.0 | -8.3 |
| LoraSeqFT | 9.2 | -24.6 |
| O-LoRA | 41.3 | **-6.2** |
| **OSFT (ours)** | **48.4** | -7.1 |
| PerTaskFT | 57.6 | NA |
| MTL | 52.3 | NA |

## 4.3 MAIN RESULTS

Table 1 shows that OSFT outperforms or matches all baselines on both 5-task and 15-task benchmarks. Importantly, compared to O-LoRA—the current SOTA parameter-efficient baseline—our method achieves superior accuracy, particularly in the more challenging 15-task scenario (71.3% vs. 69.6%), highlighting its effectiveness in maintaining task knowledge over extended task sequences. Notably, while PerTaskFT achieves high performance, it requires training separate models per task, making it computationally impractical. MTL represents an idealized scenario, training on all tasks simultaneously, thus serving as an upper-bound performance indicator. A comparison with model merging methods, SLERP and TIES, is provided in Appendix A.8, with corresponding results included in Table 1. Ablation results for rank selection and gradient projection are in Appendix A.9.

## 4.4 PERFORMANCE ON THE TRACE BENCHMARK

To further illustrate our method's capability in more realistic continual learning environments, we evaluate it on TRACE Wang et al. (2023b), which includes diverse and challenging instruction-tuning tasks.

Results in Table 2 emphasize our method's ability to effectively retain and transfer knowledge across tasks. Our approach achieves notably higher average accuracy, with slightly lower backward transfer compared to O-LoRA, demonstrating a strong balance between robustness to forgetting and adaptability to new tasks, both critical for practical deployments. In settings where tasks are related, we also observe positive transfer effects. For example, in TRACE when fine-tuning on the NumGLUE-cm and NumGLUE-ds tasks, the accuracy on NumGLUE-cm increases after training

Table 3: Comparison of general ability scores across six diverse evaluation tasks between the base LLaMA-2-7B chat model and our adaptive SVD-based continual learner.

| Model | MMLU | GSM | BBH | TydiQA | BoolQA | PIQA |
|---|---|---|---|---|---|---|
| Base Instruct Model | 46.6 | **26.1** | **40.2** | 23.5 | 70.5 | 76.2 |
| OSFT (ours) | **47.7** | 7.7 | 34.2 | **35.8** | **76.6** | **77.6** |

Table 4: Win / Tie / Lose (%) for instruction-following and safety evaluations against the LLaMA-2-7B-Chat base model.

| Method | Instruction (Helpfulness) | | | Safety | | |
|---|---|---|---|---|---|---|
| | Win | Tie | Lose | Win | Tie | Lose |
| Replay | 10 | 18 | 72 | 0 | 88 | 12 |
| LoRASeqFT | 3 | 4 | 94 | 0 | 86 | 14 |
| SeqFT | 14 | 34 | 53 | 0 | 98 | 2 |
| **OSFT (ours)** | **24** | **56** | **20** | **18** | **78** | **4** |

on NumGLUE-ds, indicating that adaptation on the later task can improve performance on the earlier, related task rather than degrade it.

**Retention of General Capabilities and Safety.** We explicitly evaluate the preservation of general abilities, instruction-following, and safety after continual learning using benchmarks proposed by TRACE. Table 3 illustrates our method's effectiveness in preserving or enhancing core language capabilities including factual knowledge, commonsense reasoning, and multilinguality compared to the original instruction-tuned model. Reasoning tasks like GSM8K suffer post-training degradation, a known issue across methods and reported in TRACE Wang et al. (2023b). Prior work highlights that continual learning without explicit reasoning supervision (e.g., chain-of-thought augmentation) is insufficient to preserve these capabilities; however, our method can be augmented with such techniques to mitigate this degradation. Table 4 demonstrates that our method also retains instruction-following ability and safety performance compared to baselines.

## 5 CONCLUSION

As large language models (LLMs) become increasingly central to real-world applications, continually adapting them without erasing prior knowledge is essential. We presented a novel continual learning framework that uses adaptive singular value decomposition (SVD) to isolate low-rank subspaces for new tasks while preserving critical directions for previously acquired knowledge. Unlike parameter-efficient techniques that freeze most weights or add modules per task, our method operates on *all* model parameters with fixed memory, preventing catastrophic forgetting through orthogonal subspace updates. Extensive empirical evaluations demonstrate our method's effectiveness across diverse benchmarks: (1) *On the 5-task benchmark with LLaMA-2 7B*, we achieved **79.6%** accuracy, surpassing the current SOTA by over 3 percentage points; (2) *or the challenging 15-task sequence with T5-Large*, we reached **71.3%** accuracy, outperforming all parameter-efficient competitors; (3) *On the realistic TRACE benchmark with LLaMA-2 7B-Chat*, our method attained **48.4%** average accuracy without requiring simultaneous multi-task access or multiple specialized models. Crucially, our approach preserved general capabilities, instruction-following behavior, and safety throughout continual learning—essential properties for deployment in production environments. Our method OSFT provides a mathematically principled solution to the fundamental tension between stability and plasticity in neural networks, offering a scalable path toward continuously evolving language models that efficiently accumulate knowledge without forgetting. Our work establishes a practical approach for real-world deployment of continually adapting language models. Limitations and future directions are discussed in Appendix A.2.

REPRODUCIBILITY STATEMENT

We provide all details needed to reproduce our results. The algorithm is specified in Section 3 with the training procedure summarized in Algorithm 1. Benchmarks, metrics, and evaluation protocol are described in Section 4. Theoretical assumptions and full proofs are provided in Appendix A.3. Computational cost and memory analysis are in Appendix A.11, with ablations in Appendix A.9. Per-task results and task orders are reported in Appendix A.12. Implementation and hyperparameters are given in Appendix A.10. A code repository with scripts and configs is linked in Appendix A.13. Our disclosure of LLM usage is in Appendix A.14.

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

# A  APPENDIX

## A.1  ALGORITHM SUMMARY

---

**Algorithm 1** Orthogonal Subspace Fine-Tuning (OSFT) - Practical Implementation

---

1: **Require:** Initial parameters $\theta = \{\mathbf{W}^{(l)}\}_{l=1}^{L}$, tasks $\{\mathcal{D}_t\}_{t=1}^{T}$, hyperparameters $\mathrm{mrr}, \mathrm{trr}$.
2: **Ensure:** Parameters are updated continually while preserving high-rank subspaces.
3: **for** task $t = 1, \ldots, T$ **do**
4:     Compute layer importance $I^{(l)}$ and determine retention count $k^{(l)}$.
5:     **for** layer $l = 1, \ldots, L$ **do**
6:         Decompose $\mathbf{W}^{(l)}$ via SVD.
7:         **Reparameterize Layer:**
8:             Store high-rank components $(\mathbf{U}_{\mathrm{high}}^{(l)}, \ldots)$ as frozen buffers.
9:             Store low-rank components $(\mathbf{U}_{\mathrm{low}}^{(l)}, \ldots)$ as new trainable parameters.
10:         Register a gradient hook on the trainable SVD components to enforce orthogonality.
11:     **end for**
12:     **while** not converged on task $\mathcal{D}_t$ **do**
13:         Sample mini-batch.
14:         In the forward pass, reconstruct effective weight $\mathbf{W}_{\mathrm{eff}}^{(l)} = \mathbf{W}_{\mathrm{high}}^{(l)} + \mathbf{W}_{\mathrm{low}}^{(l)}$.
15:         Backward pass computes gradients for trainable SVD components (e.g., $\nabla \mathbf{U}_{\mathrm{low}}^{(l)}$).
16:         The gradient hook automatically projects these gradients.
17:         The optimizer updates only the (projected) trainable SVD components.
18:     **end while**
19: **end for**

---

## A.2  LIMITATIONS AND FUTURE WORK

While our approach achieves strong performance across a range of benchmarks, few directions remain open for further refinement. **(1) Rank Estimation Heuristics:** Although our current rank selection method performs robustly in practice, future work could explore more principled, data-driven heuristics to fine-tune retention ratios with even greater precision. **(2) Layer-Specific Optimization:** Our method currently applies SVD to all weight matrices; selectively targeting specific layer types (e.g., attention matrices) may offer further efficiency gains with minimal trade-offs. **(3) Long-Horizon Adaptation:** In scenarios with very large numbers of tasks, more adaptive capacity management or online adjustment of subspace budgets may further enhance scalability. **(4) Samples from Previous Task:** When we use the cosine-similarity based rank selection to set the effective rank, OSFT requires a small fixed-size buffer of task $(t-1)$ data when starting task $t$; this is a fixed memory requirement and does not increase with the number of tasks in the sequence. The alternative predetermined-threshold schedule for effective rank does not require any old-task data and also performs well in practice. These are natural extensions to our core framework, which remains effective and practical in current continual learning settings.

## A.3  THEORETICAL ANALYSIS: TIGHTER FORGETTING BOUNDS VIA ADAPTIVE SVD

We now formally derive a hierarchy of catastrophic forgetting bounds that rigorously demonstrate the advantage of our adaptive rank selection approach compared to both naive full fine-tuning and uniform low-rank projection methods. *In essence, this section shows how protecting high-curvature directions (i.e., large Hessian eigenvalues) minimizes forgetting—motivating our subsequent use of weight-matrix SVD as a tractable approximation.*

**Lemma 1** (Second-Order Approximation of Catastrophic Forgetting)**.** *Consider a model with parameters $\theta^{(k)}$ after training on task $k$, and subsequent parameters $\theta^{(k+1)} = \theta^{(k)} + \Delta\theta$ after learning task $k+1$. Assuming $\nabla L_k(\theta^{(k)}) \approx 0$ (i.e., task $k$'s loss is near-optimal at $\theta^{(k)}$), the catastrophic forgetting on task $k$ can be approximated by:*

$$\Delta L_k \triangleq L_k(\theta^{(k+1)}) - L_k(\theta^{(k)}) \approx \frac{1}{2}\Delta\theta^{\top} H_k \Delta\theta, \tag{6}$$

where $H_k = \nabla^2 L_k(\theta^{(k)})$ is the Hessian of the loss function at $\theta^{(k)}$.

*Proof.* **Step 1: Taylor Expansion.** Expanding $L_k$ at $\theta^{(k+1)} = \theta^{(k)} + \Delta\theta$ via Taylor's theorem:

$$L_k(\theta^{(k+1)}) = L_k(\theta^{(k)}) + \underbrace{\nabla L_k(\theta^{(k)})^\top \Delta\theta}_{\approx 0} + \frac{1}{2}\Delta\theta^\top H_k \Delta\theta + O(\|\Delta\theta\|^3). \quad (7)$$

**Step 2: First-Order Term Vanishes.** Since $\theta^{(k)}$ represents a (local) optimum for task $k$, we have $\nabla L_k(\theta^{(k)}) \approx \mathbf{0}$, thereby eliminating the first-order term.

**Step 3: Dominant Quadratic Term.** The remaining quadratic term $\frac{1}{2}\Delta\theta^\top H_k \Delta\theta$ dominates forgetting. $\qquad\square$

**Lemma 2** (Block-Diagonal Approximation of the Hessian). *Consider a Transformer model with parameters partitioned into layers such that:*

$$\theta = \left[\text{vec}(W^{(1)})^\top, \text{vec}(W^{(2)})^\top, \dots, \text{vec}(W^{(L)})^\top\right]^\top.$$

*The Hessian matrix $H_k$ at the optimum $\theta^{(k)}$ can be approximated as block-diagonal with respect to layers:*

$$H_k \approx \begin{bmatrix} H_k^{(1)} & 0 & \cdots & 0 \\ 0 & H_k^{(2)} & \cdots & 0 \\ \vdots & \vdots & \ddots & \vdots \\ 0 & 0 & \cdots & H_k^{(L)} \end{bmatrix}, \quad (8)$$

*where each $H_k^{(\ell)}$ represents the intra-layer Hessian for layer $\ell$. Under this approximation, the quadratic form decomposes as:*

$$\Delta\theta^\top H_k \Delta\theta \approx \sum_{\ell=1}^{L} \text{vec}(\Delta W^{(\ell)})^\top H_k^{(\ell)} \text{vec}(\Delta W^{(\ell)}). \quad (9)$$

*Proof.* The block-diagonal approximation is theoretically justified by analyses showing the Hessian of neural networks, especially Transformers, is dominated by intra-layer terms with negligible cross-layer interactions (Singh et al., 2021; Martens & Grosse, 2015). Empirical evidence from Transformer models further supports this structure: Hessian spectrum analyses reveal minimal magnitude in off-diagonal inter-layer Hessian blocks compared to the intra-layer blocks (Zhang et al., 2024).

**Empirical Validation:** As shown in Zhang et al. (2024), inter-layer Hessian blocks in Transformers exhibit $\sim 10\times$ smaller Frobenius norms than intra-layer blocks, with cross-layer correlations below $0.1$ in pretrained models. This justifies treating layers independently for curvature analysis.

**Norm Equivalence:** Note that $\text{vec}(\Delta W^{(\ell)})^\top H_k^{(\ell)}\text{vec}(\Delta W^{(\ell)})$ is equivalent to $\langle \Delta W^{(\ell)}, H_k^{(\ell)}\Delta W^{(\ell)}\rangle_F$, where $\langle\cdot,\cdot\rangle_F$ is the Frobenius inner product. Thus, the quadratic form directly ties to layer-wise Frobenius norms.

In practice, optimization and continual learning algorithms that assume a block-diagonal Hessian, such as Kronecker-Factored Approximate Curvature (K-FAC) (Martens & Grosse, 2015) and structured Laplace approximations (Ritter et al., 2018), consistently demonstrate effectiveness in leveraging layer-wise curvature without significant loss of accuracy. Thus, the approximation is both theoretically sound and empirically validated. $\qquad\square$

**Lemma 3** (Relationship Between Layer Importance and Curvature). *The layer importance measure $I^{(\ell)}$, defined as:*

$$I^{(\ell)} = \frac{1}{N}\sum_{i=1}^{N} cosine\_similarity(X_i^{(\ell)}, Y_i^{(\ell)}) \quad (10)$$

*where $X_i^{(\ell)}$ are layer inputs and $Y_i^{(\ell)} = W^{(\ell)}X_i^{(\ell)}$ are layer outputs, positively correlates with the spectral properties of the layer-wise Hessian $H_k^{(\ell)}$.*

*Proof.* Layers with high importance scores (high similarity between inputs and outputs) tend to preserve activation patterns rather than significantly transform them. These layers typically serve as information conduits in the network, maintaining critical features learned for task $k$.

Empirically, these high-importance layers exhibit higher sensitivity to parameter perturbations. When a layer primarily passes information forward with minimal transformation (high $I^{(\ell)}$), perturbations to its parameters directly interfere with this information flow, causing large changes in the loss function. Mathematically, this translates to larger eigenvalues in $H_k^{(\ell)}$, indicating steeper curvature.

Conversely, layers with lower $I^{(\ell)}$ values significantly transform their inputs, suggesting these layers are more adaptable. Perturbations to these layers' parameters cause smaller changes in the loss landscape, resulting in smaller eigenvalues in $H_k^{(\ell)}$.

This relationship has been verified empirically in multiple studies (Sharma et al., 2023; Li et al., 2025), consistently showing a positive correlation between measures of layer importance and the magnitude of Hessian eigenvalues.

**Intuition:** Consider a layer that merely passes input features (high $I^{(\ell)}$). Perturbing its weights $W^{(\ell)}$ directly distorts critical task-$k$ features, causing large loss changes (high curvature). In contrast, layers transforming inputs (low $I^{(\ell)}$) allow parameter changes without catastrophic feature distortion, corresponding to flatter curvature. $\square$

**Preserving Large Hessian Eigenvalues Minimizes Forgetting.** Combining these lemmas, we see that *directions with large Hessian eigenvalues* impose the greatest risk for catastrophic forgetting: even small updates along those directions yield substantial loss increases for old tasks.

**Theorem 1** (Hierarchy of Forgetting Bounds). *Assuming equal parameter update magnitudes $\|\Delta\theta\|^2 = c$ across different fine-tuning strategies, the forgetting bounds satisfy:*

$$\text{Adaptive SVD} < \text{Fixed-Rank} < \text{Full Fine-tuning} \tag{11}$$

*Specifically:*

$$\text{Full Fine-tuning:} \quad \Delta L_k \le \frac{1}{2}\lambda_{\max}(H_k) \cdot c, \tag{12}$$

$$\text{Fixed-rank:} \quad \Delta L_k \le \frac{1}{2}\max_{\ell}\{\lambda_{r+1}^{(\ell)}\} \cdot c, \tag{13}$$

$$\text{Adaptive (Ours):} \quad \Delta L_k \le \frac{1}{2}\max_{\ell}\{\lambda_{r(\ell)+1}^{(\ell)}\} \cdot c, \tag{14}$$

*where $r(\ell) = mrr + I^{(\ell)}(trr - mrr)$ is our adaptive rank allocation based on layer importance.*

*Moreover, under the condition that layer importance $I^{(\ell)}$ positively correlates with Hessian curvature (Lemma 3), we have:*

$$\max_{\ell}\{\lambda_{r(\ell)+1}^{(\ell)}\} < \max_{\ell}\{\lambda_{r+1}^{(\ell)}\} \le \lambda_{\max}(H_k), \tag{15}$$

*ensuring our adaptive approach provides strictly tighter forgetting bounds.*

*Proof.* We establish the hierarchy of bounds by proving each inequality separately.

**Part 1:** $\max_{\ell}\{\lambda_{r+1}^{(\ell)}\} \le \lambda_{\max}(H_k)$. By the block-diagonal approximation (Lemma 2), $\lambda_{\max}(H_k) = \max_{\ell}\{\lambda_1^{(\ell)}\}$. From Lemma 3, high-$I^{(\ell)}$ layers have larger $\lambda_1^{(\ell)}$. Since $\lambda_{r+1}^{(\ell)} \le \lambda_1^{(\ell)}$ for all $\ell$ by the ordering of eigenvalues, we have:

$$\max_{\ell}\{\lambda_{r+1}^{(\ell)}\} \le \max_{\ell}\{\lambda_1^{(\ell)}\} = \lambda_{\max}(H_k).$$

**Rayleigh Quotient Proof for Full Fine-tuning Bound:** For the full fine-tuning case, we need to bound $\Delta\theta^\top H_k \Delta\theta$. By the Rayleigh quotient property, for any symmetric matrix $H_k$ and non-zero vector $\Delta\theta$:

$$\frac{\Delta\theta^\top H_k \Delta\theta}{\|\Delta\theta\|^2} \le \lambda_{\max}(H_k),$$

where $\lambda_{\max}(H_k)$ is the largest eigenvalue of $H_k$. This holds because the maximum value of the Rayleigh quotient equals the largest eigenvalue.

Rearranging, we get:

$$\Delta\theta^\top H_k \Delta\theta \;\leq\; \lambda_{\max}(H_k) \cdot \|\Delta\theta\|^2 \;=\; \lambda_{\max}(H_k) \cdot c.$$

Hence the forgetting bound for full fine-tuning is:

$$\Delta L_k \approx \tfrac{1}{2} \Delta\theta^\top H_k \Delta\theta \;\leq\; \tfrac{1}{2} \lambda_{\max}(H_k) \|\Delta\theta\|^2.$$

**Part 2:** $\max_\ell\{\lambda^{(\ell)}_{r(\ell)+1}\} < \max_\ell\{\lambda^{(\ell)}_{r+1}\}$.

Let $\ell^* = \arg\max_\ell \lambda^{(\ell)}_{r+1}$ be the layer with the largest post-projection eigenvalue in the fixed-rank approach. By Lemma 3, this layer typically has high curvature and thus high importance $I^{(\ell^*)}$. Under our adaptive allocation strategy, that high-importance layer obtains a larger rank allocation ($r(\ell^*) > r$), ensuring:

$$\lambda^{(\ell^*)}_{r(\ell^*)+1} < \lambda^{(\ell^*)}_{r+1} = \max_\ell\{\lambda^{(\ell)}_{r+1}\}.$$

For any other layer $\ell \neq \ell^*$,

$$\lambda^{(\ell)}_{r(\ell)+1} < \lambda^{(\ell^*)}_{r+1} = \max_\ell\{\lambda^{(\ell)}_{r+1}\},$$

either because $r(\ell) > r$ (for other high-importance layers) or because $\lambda^{(\ell)}_{r+1} < \lambda^{(\ell^*)}_{r+1}$ (for low-importance layers). Hence $\max_\ell\{\lambda^{(\ell)}_{r(\ell)+1}\} < \max_\ell\{\lambda^{(\ell)}_{r+1}\}$, implying a strictly tighter bound than fixed-rank.

Combining Parts 1 and 2 completes the proof of the bound hierarchy. $\square$

$$\underbrace{\lambda^{(\ell^*)}_{r(\ell^*)+1}}_{\text{Adaptive (Ours)}} \;<\; \underbrace{\lambda^{(\ell^*)}_{r+1}}_{\text{Fixed-Rank}} \;\leq\; \underbrace{\lambda^{(\ell^*)}_{1}}_{\text{Full Fine-Tuning}} \;, \tag{16}$$

where $\ell^* = \arg\max_\ell \lambda^{(\ell)}_{r+1}$ is the highest-curvature layer.

**On the Equal-Norm Assumption** The assumption $\|\Delta\theta\|^2 = c$ across different fine-tuning strategies isolates the impact of update directions but does not imply optimality. In practice:

- Adaptive SVD may achieve lower forgetting *even with smaller norms* by avoiding high-curvature directions.
- Full fine-tuning could offset poor directional alignment with larger updates, but this risks catastrophic forgetting.
- Future work should analyze the Pareto frontier of the accuracy–forgetting trade-off under variable norms.

This assumption is purely a theoretical device, not a claim about how hyperparameters are tuned in practice.

> **Key Theoretical Insights**
>
> Under equal parameter update budgets:
>
> - Full fine-tuning suffers worst-case forgetting bounded by $\lambda_{\max}(H_k)$.
> - Fixed-rank projection improves on this by capping directions via a uniform low-rank selection, but misallocates rank to some layers.
> - Adaptive SVD aligns per-layer rank $r(\ell)$ with curvature (via $I^{(\ell)}$), giving strictly tighter forgetting bounds.

**Corollary 1** (Forgetting Reduction with Adaptive SVD). *Under the equal parameter update magnitude assumption, our adaptive SVD achieves strictly less forgetting than fixed-rank or naive full fine-tuning. This gap widens when:*

- *Layer importance $I^{(\ell)}$ varies significantly across layers,*

- *The Hessian spectrum shows heavy tails (a few large eigenvalues dominate).*

*Proof.* Follows directly from Theorem 1 and the established bound hierarchy:

$$\Delta L_k^{\text{Adaptive}} \;<\; \Delta L_k^{\text{Fixed-rank}} \;<\; \Delta L_k^{\text{Full}}.$$

$\square$

**Practical Approximation via Weight-Matrix SVD.** While the above results show that *retaining large Hessian-eigenvalue directions* is essential to minimize forgetting, *computing* Hessian eigenvectors is intractable for large language models. Recent empirical findings (Haink, 2023) indicate that these high-curvature directions often overlap significantly with top singular vectors of the weight matrices. Hence, our method uses SVD-based rank selection—preserving large singular values—as a pragmatic surrogate for preserving large Hessian eigenvalues. By focusing on lower singular-value directions for new-task updates, we effectively contain catastrophic forgetting without the prohibitive overhead of Hessian decomposition. This aligns with the theoretical ideal of limiting updates where curvature is highest, but in a computationally feasible manner.

This theoretical framework underpins our *adaptive* SVD strategy: high-importance layers (with higher curvature) get more singular directions retained, while less critical layers can be more aggressively pruned. As shown in Section 4, this approach consistently outperforms naive full fine-tuning and uniform low-rank baselines in mitigating forgetting and stabilizing knowledge across tasks.

A.4    EMPIRICAL VALIDATION OF LOW RANK APPROXIMATION

We conducted an in-depth analysis of the Granite 8B model architecture to validate findings from prior literature suggesting that the weight matrices in transformer layers are effectively low-rank (Sharma et al., 2023; Hartford et al., 2024). This implies that these matrices can be accurately approximated using low-rank Singular Value Decomposition (SVD), revealing unused capacity that can potentially be leveraged to learn additional tasks or improve performance on existing ones. Since Granite shares a similar architecture with LLaMA, our findings are directly applicable to LLaMA and offer broader insights into decoder-only transformer architectures and large language models in general.

Table 5: Leaderboard average results for `attn.k_proj.weight` across varying low-rank reduction levels. Middle layers showed slightly better robustness than early layers. The baseline here refers to the original Granite 8B model without any low-rank approximation.

| Reduction % | Above Baseline | Below Baseline |
|---|---|---|
| 10% | 3 | 9 |
| 50% | 4 | 8 |
| 90% | 2 | 10 |
| 99% | 2 | 10 |
| 99.75% | 0 | 11 |

We examined all attention and feedforward projection matrices across all layers of Granite 8B, and report results for four key matrices: the attention value and key projections, and the two feedforward projection matrices that follow attention. Based on prior observations from LASER Sharma et al. (2023) suggesting that later layers benefit most from rank reduction—often leading to improved downstream performance when high-frequency components are removed—we report findings from layers 28, 29, 34, and 39 out of the model's 40 layers. We performed SVD-based low-rank approximations at varying reduction levels (e.g., retaining only 1%, 50%, or 90% of the original singular

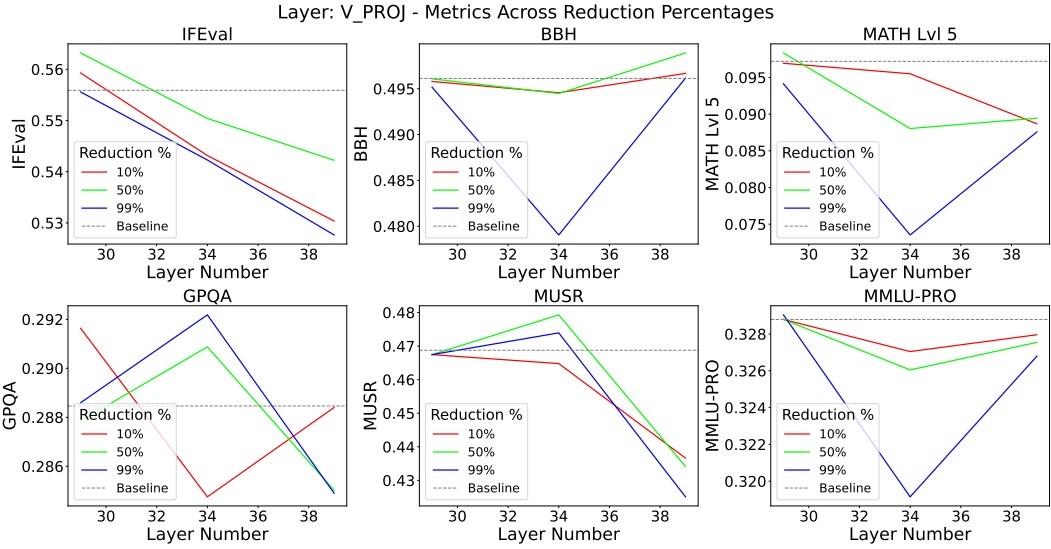

Figure 3: Effect of low-rank approximation on the `attn.v_proj.weight` (value projection matrix) across selected layers in Granite 8B, evaluated on the Leaderboard benchmark.

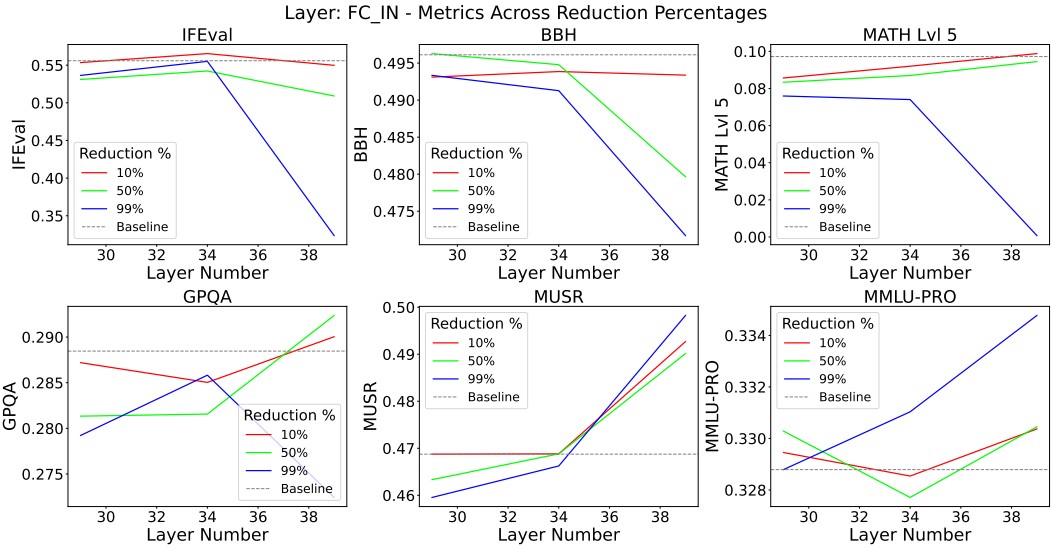

Figure 4: Effect of low-rank approximation on the `mlp.gate_proj.weight` (first feedforward projection) across selected layers in Granite 8B, evaluated on the Leaderboard benchmark.

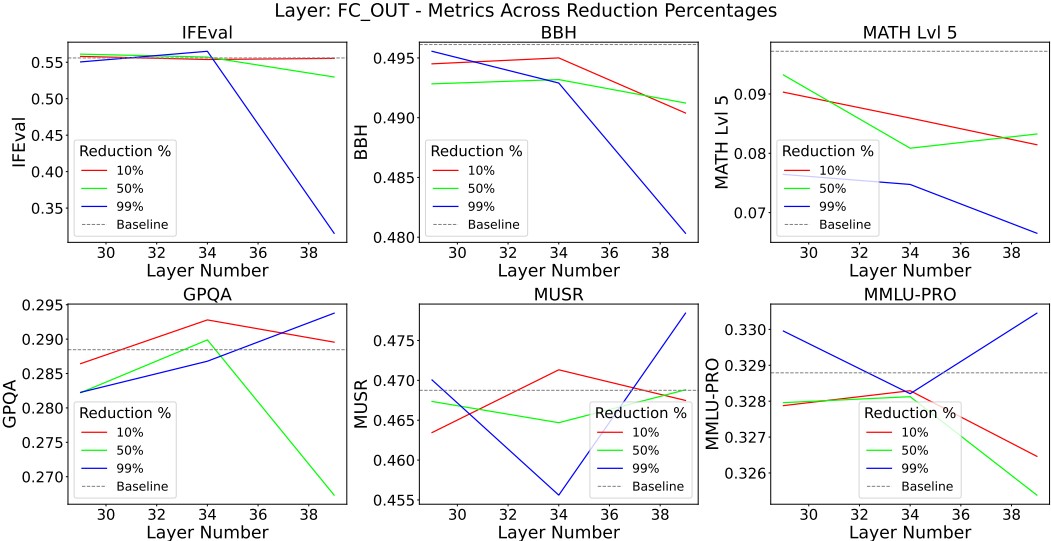

Figure 5: Effect of low-rank approximation on the `mlp.down_proj.weight` (third feedforward projection) across selected layers in Granite 8B, evaluated on the Leaderboard benchmark.

vectors), and evaluated the impact of each intervention on performance on the Open LLM Leaderboard v2 benchmark[1] consisting of six tasks — MMLU-Pro, GPQA, MuSR, MATH, IFEval, and BBH. Consistent with prior work, we observed that some low-rank approximations maintained or even improved performance, highlighting the redundancy and compressibility of these matrices (see Figures 3, 4, and 5 and Table 5). Each experiment involved a single intervention defined by a tuple specifying the layer number, matrix type, and reduction percentage.

Our approach assumes that lower singular vectors can safely accommodate new knowledge without significant forgetting. Specifically, our method relies on the premise that fine-tuning in the directions of low singular vectors will not interfere with previously learned tasks. This assumption holds only if the data from earlier tasks lie predominantly in the subspace spanned by the high singular vectors. If task-specific information from earlier tasks resides in the span of the low singular vectors, modifying these directions could lead to interference—especially if the associated singular values were previously small (effectively suppressing higher-frequency components or noise), but are increased during learning on new tasks, thereby reactivating those suppressed directions. Formally, we expand the weight matrix via SVD as:

$$\mathbf{W} = \sum_{i=1}^{r} \sigma_i \, \mathbf{u}_i \mathbf{v}_i^\top \tag{17}$$

To empirically verify this, we investigate whether the output components of previous tasks in the hidden layer, when projected onto the low singular vector subspace, are negligible. In particular, we compute the L2 norm of the matrix-vector product between the outer product of each singular vector pair $\mathbf{u}_i \mathbf{v}_i^\top$ and the input vector (from a previously learned task) without scaling by the corresponding singular value. This helps determine whether the old task input lies in the null space of the low singular vectors or merely yield small outputs due to low singular values. If the L2 norms of the matrix-vector products corresponding to low singular vectors are near zero, we can safely update these directions for new tasks without affecting the prior task.

We perform this analysis on the `mlp.down_proj.weight` matrix in layer 34 of Granite 8B using data from a previously learned task. The results are presented in Figure 6. As expected, the output norm steadily decreases from left to right, where the x-axis corresponds to singular vector indices sorted in descending order of singular values. The three highest singular directions yield

---

[1]https://huggingface.co/docs/leaderboards/open_llm_leaderboard/about

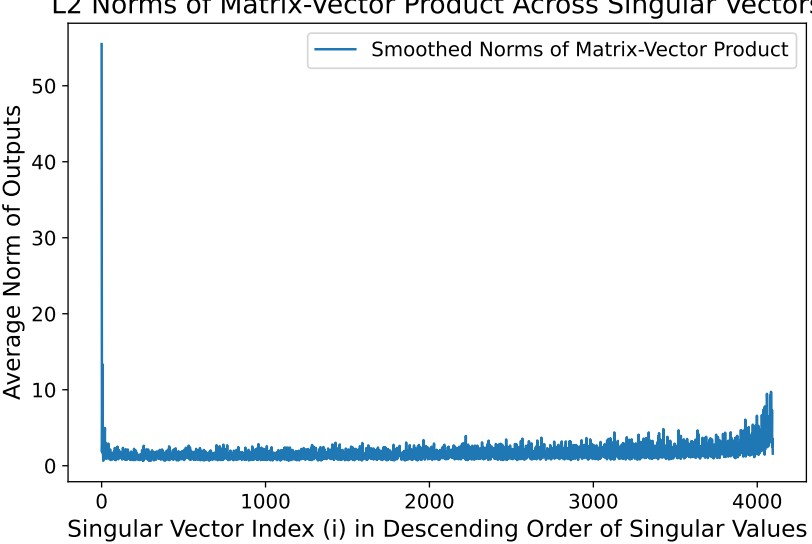

Figure 6: L2 norms of matrix-vector products for each singular vector component in the `mlp.down_proj.weight` matrix (layer 34, Granite 8B), using inputs from a previously learned task. The clear downward trend confirms that low singular directions have minimal activation for the learned task.

norms of 55.5, 18.1 and 1.8, respectively, indicating a sharp drop in signal strength after the top components. This supports the theoretical redundancy hypotheses (Chen et al., 2020; Sharma et al., 2023), validating our adaptive low-rank continual learning strategy. In particular, this layer retained performance even after a 99% rank reduction, matching the performance of the unmodified Granite 8B model on the Leaderboard benchmark.

These diagnostic experiments laid the groundwork for our final approach, which leverages projected gradient descent restricted to low-rank subspaces. Importantly, these subspaces are adaptively selected to minimize interference with previously learned tasks while preserving expressive capacity for learning new ones. Detailed analysis of singular value statistics across all layers and matrix types is provided in Appendix A.5.

## A.5 SINGULAR VALUE STATISTICS AND RANK ANALYSIS OF THE GRANITE 8B MODEL

To better understand how to select which singular vectors to fine-tune within model weight matrices, we analyzed the singular value statistics of each matrix using tools from Random Matrix Theory (RMT). Specifically, we examined the use of the lower bound of the Marchenko–Pastur distribution—following the approach in SPECTRUM (Hartford et al., 2024)—to distinguish signal from noise. Singular values that fell below this bound were treated as noise, allowing us to estimate the effective rank of each matrix. However, we observed that, under this criterion, all weight matrices in the Granite 8B model appear to be full-rank. This outcome is attributed to the violation of the core assumptions of the Marchenko–Pastur law—namely, that matrix entries are independently and identically distributed—which clearly does not hold in trained language models where parameters are highly structured and correlated. Consequently, we adopted a scaled thresholding approach, informed by descriptive statistics such as the minimum, mean, median, and maximum singular values within each layer.

To support the adaptive rank selection strategy introduced in the main paper, we performed a comprehensive analysis of the singular value spectra across all weight matrices in the Granite 8B model. For each matrix type (e.g., `q_proj`, `k_proj`, `v_proj`, `o_proj`, `gate_proj`, `up_proj`, `down_proj`), we compute and visualize the distribution of minimum, maximum, mean, and median singular values across all transformer layers (Figures 7–13). We also construct a heatmap illustrating the variation of mean singular values throughout the network (Figure 14). These statistics provide

useful insights into which low singular vectors and corresponding subspaces are suitable for fine-tuning during continual learning.

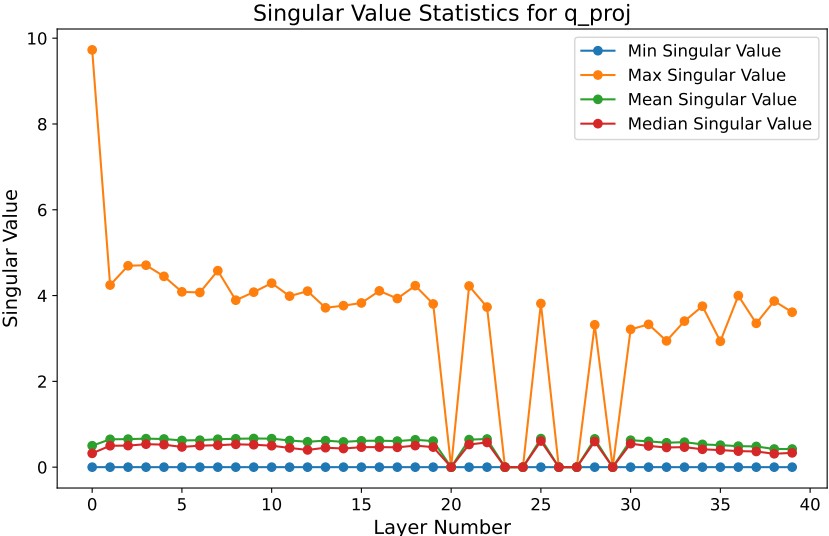

Figure 7: Singular value statistics for the `attn.q_proj.weight` matrix across Granite 8B layers.

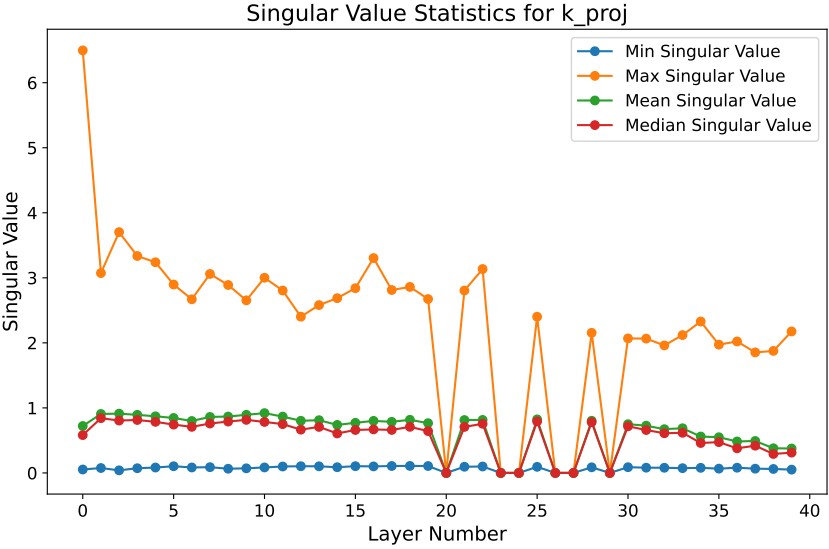

Figure 8: Singular value statistics for the `attn.k_proj.weight` matrix across layers.

### A.6 COMPARISON WITH SVD BASELINES: MILORA AND PISSA

MiLoRA and PiSSA are closely related to our setting, as they also use SVD to decompose weight matrices and then update only a subset of singular directions using LoRA-style adapters. MiLoRA performs SVD of the weight matrix, freezes the high-singular-value components (interpreted as the "pretrained base"), and updates the low-singular-value components as trainable adapters. PiSSA applies the opposite split: it freezes the low-singular-value components and updates the high-singular-value components as adapters.

For a direct comparison, we follow the math reasoning setup from the MiLoRA paper and fine-tune LLaMA-2-7B on the MetaMathQA dataset (395K samples combining GSM8K and MATH). We

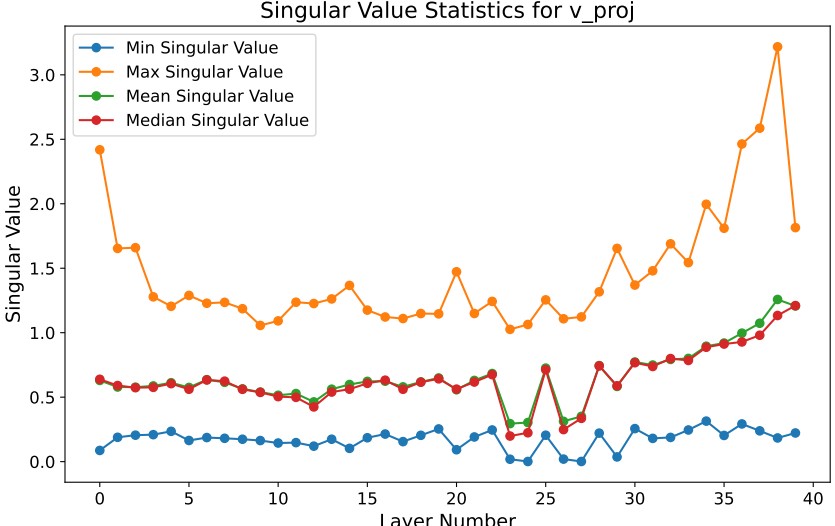

Figure 9: Singular value statistics for the `attn.v_proj.weight` matrix across layers.

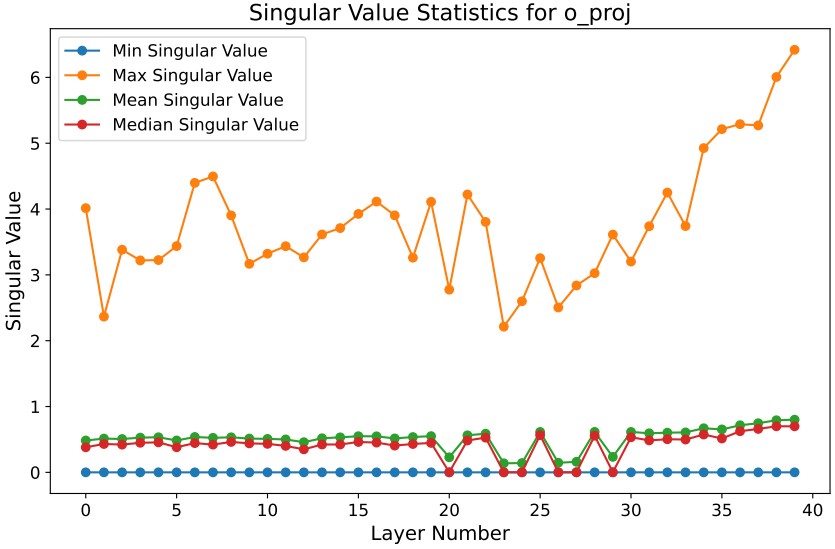

Figure 10: Singular value statistics for the `attn.o_proj.weight` matrix across layers.

evaluate on the GSM8K and MATH test sets and report Exact Match (EM) on the final checkpoint. Table 6 summarizes the results.

Table 6: Comparison with MiLoRA and PiSSA on the MetaMathQA setup using LLaMA-2-7B. We report Exact Match (EM) on GSM8K and MATH, along with the average.

| Method | GSM8K | MATH | Avg. |
|---|---|---|---|
| Full FT | 66.5 | **19.8** | 43.2 |
| LoRA | 60.6 | 16.9 | 38.7 |
| PiSSA | 58.2 | 15.8 | 37.0 |
| MiLoRA | 63.5 | 17.8 | 40.7 |
| **OSFT (ours)** | **69.7** | 18.2 | **43.95** |

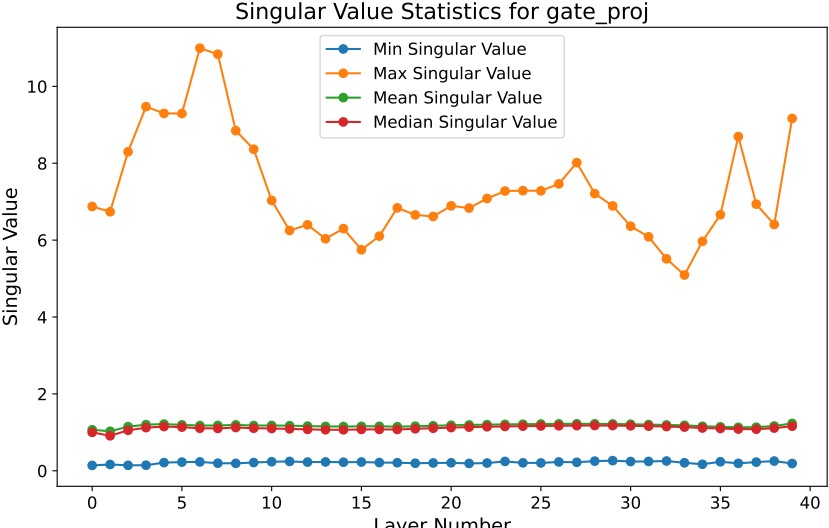

Figure 11: Singular value statistics for the `mlp.gate_proj.weight` matrix across layers.

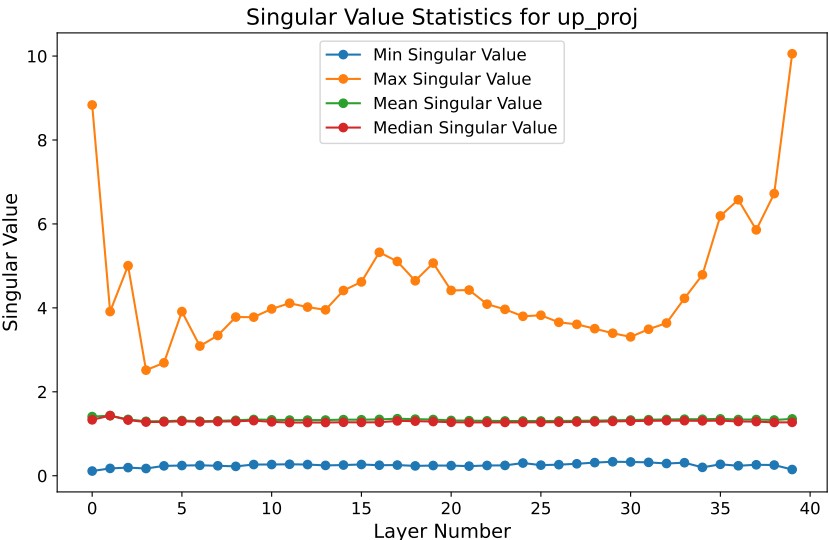

Figure 12: Singular value statistics for the `mlp.up_proj.weight` matrix across layers.

OSFT slightly improves on the full fine-tuning average and outperforms both MiLoRA and PiSSA on this math reasoning setup, indicating that constraining updates to an orthogonal low-singular-value subspace can match or exceed strong SVD-based PEFT baselines while maintaining the continual learning benefits studied in the main paper.

### A.7 COMPARISON WITH SPARSE FINE-TUNING

Sparse fine-tuning approaches aim to mitigate forgetting by selectively freezing or reactivating important parameters. We compare our method against Lottery Ticket Adaptation (LoTA) (Panda et al., 2024), which uses sparsity masks to preserve critical weights from a source task while adapting to a new task. We reproduce the task transfer setup from Table 5 of their paper using the Mistral-7B-v0.1 model. Each experiment begins with an instruction-following task (Task A), followed by transfer to a downstream task (Task B). The goal is to preserve performance on Task A while adapting effectively to Task B.

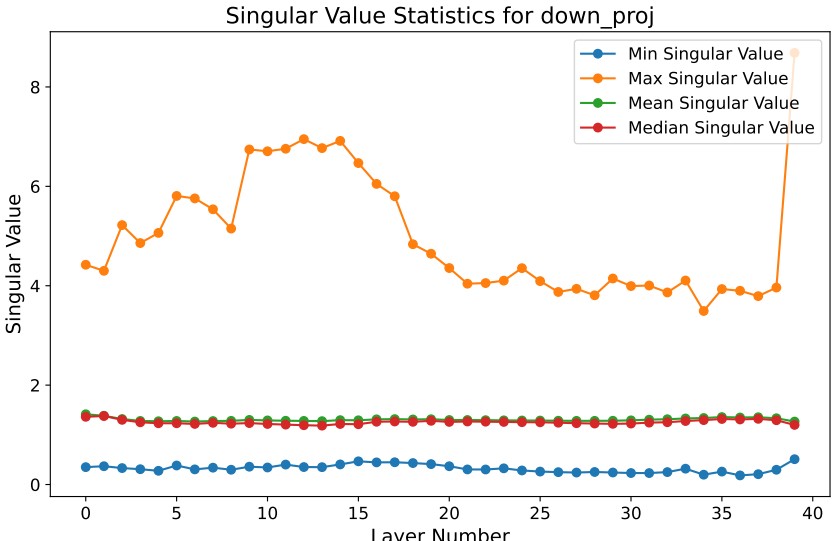

Figure 13: Singular value statistics for the `mlp.down_proj.weight` matrix across layers.

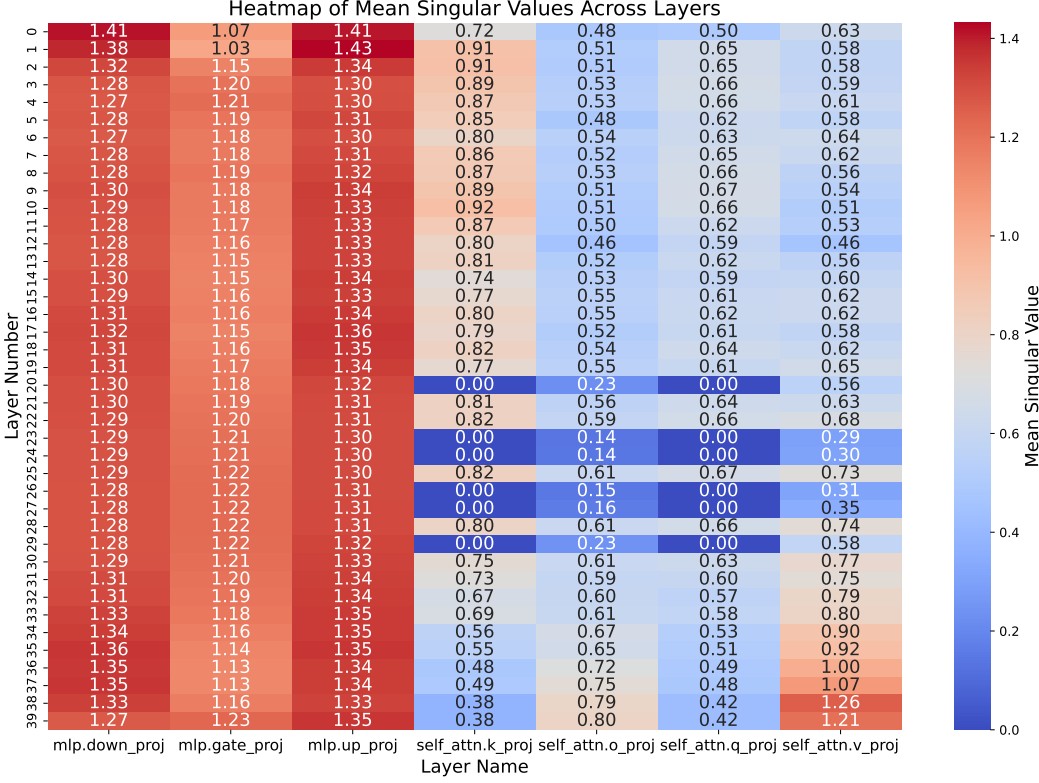

Figure 14: Heatmap of mean singular values across all matrices and transformer layers in Granite 8B.

As shown in Table 7, our method OSFT consistently matches or outperforms LoTA in mitigating forgetting (Task A) while achieving comparable or higher downstream task performance (Task B). For example, on GSM8K and MathInstruct, our method achieves stronger retention and higher accuracy

Table 7: Comparison between LoTA and our method on sequence of two tasks. Each row corresponds to a setup where the model is first trained on the instruction-following task (Task A) and then fine-tuned on a downstream task (Task B). We report the utility (accuracy or task-specific metric) on both Task A and Task B after training on the second task. Values in parentheses indicate the drop in Task A or Task B performance compared to the **Baseline**, where the model is trained on that task alone. Lower drop in Task A utility indicates better forgetting mitigation; higher Task B utility indicates better learning and adaptation.

| Task B | Task A Method | Task B Method | Task A Utility ($\downarrow$) | Task B Utility ($\downarrow$) |
|---|---|---|---|---|
| Instruction Follow | Baseline | - | 19.0 (-) | - |
| GSM8K | - | Baseline | - | 59.8 (-) |
| | FFT | OSFT (ours) | **18.6 (0.4)** | **60.1 (-)** |
| | LoTA | LoTTO | 17.8 (1.2) | 59.1 (0.7) |
| MathInstruct | - | Baseline | - | 56.7 (-) |
| | FFT | OSFT (ours) | **17.8 (1.2)** | **60.1 (-)** |
| | LoTA | LoTA | 16.0 (3.0) | 55.5 (1.2) |
| Reasoning | - | Baseline | - | 83.5 (-) |
| | FFT | OSFT (ours) | **18.3 (0.7)** | 82.1 (1.4) |
| | LoTA | LoTTO | 16.5 (2.5) | **83.7 (-)** |
| GSM8K+Arc+SQL | - | Baseline | - | 77.0 (-) |
| | FFT | OSFT (ours) | 14.7 (4.3) | **76.2 (0.8)** |
| | LoTA | LoTTO | **15.9 (3.1)** | 73.8 (3.2) |
| Safety | Baseline | - | 93.1 (-) | - |
| | FFT | OSFT (ours) | **71.8 (21.3)** | - |
| | LoTA | LoTTO | 63.4 (29.7) | - |

on the new task. On Reasoning, we maintain near-zero drop in Task A utility, while LoTA suffers a larger decline. These results demonstrate the effectiveness of the full-parameter constrained update approach relative to sparse masking.

## A.8 COMPARISON WITH MODEL MERGING TECHNIQUES

We compare against two model merging techniques—SLERP (Spherical Linear Interpolation) and TIES (Task-Informed Ensemble Synthesis)—to assess their applicability in the continual learning setting. SLERP was applied by merging full model weights sequentially: after each task, the model was interpolated with the next task's model on the unit hypersphere. TIES was applied to linearly combine task-specific LoRA adapters using weights tuned on a held-out validation set. Our method OSFT significantly outperforms both (see Table 1). In continual learning benchmarks involving many tasks, such as the 5-task and 15-task settings examined here, finding effective merge strategies becomes increasingly challenging. Moreover, even after identifying an optimal strategy, extensive hyperparameter tuning, experimentation, and expert knowledge are typically required to merge models effectively without compromising task performance over long task sequences. This complexity makes such merging approaches less practical compared to our proposed method.

## A.9 ABLATION STUDIES

To better understand the contribution of key components in our method, we conduct two ablation studies using the LLaMA-2 7B model on the standard continual learning benchmark comprising 5 classification tasks (AG News, Amazon, Yelp, DBpedia, Yahoo). These ablations are designed to evaluate: (1) the importance of accurate effective rank estimation for singular vector selection, and (2) the necessity of constraining updates to remain within the low-rank subspace via projection.

**(1) Impact of Inaccurate Effective Rank Estimation:** Our method relies on computing an effective rank per matrix based on input-output activation similarity, which informs the threshold for partitioning singular vectors into high- and low-rank subspaces. To test the importance of this estimation, we reduce both the minimum and target retention ratios (mrr and trr) to half their original

values. This results in more aggressive fine-tuning by retaining fewer high singular vectors, thus allocating more of the matrix capacity to learning new tasks. However, this also increases the risk of overwriting components important for previous tasks. As shown in Table 9, this ablation leads to a substantial performance drop of just over 28 percentage points (from 79.6% to 51.5%), emphasizing the importance of accurately estimating the effective rank to ensure that task-relevant subspaces are preserved.

Ablation results over mrr and trr are shown in Table 8. The default setting $(0.10, 0.80)$ gives the best average accuracy. Nearby values such as $(0.05, 0.70)$ and $(0.20, 0.90)$ perform similarly (within $\sim$4–8 points), showing that OSFT is reasonably robust to moderate changes. Very low retention $(0.05, 0.40)$ and "flat" schedules where mrr $=$ trr (e.g., $(0.50, 0.50)$) hurt performance, and the extreme retention $(0.70, 1.00)$ case performs worst, confirming that overly weak retention or overly high retention both degrade results.

Table 8: Sensitivity of OSFT to mrr and trr on the 5-task standard continual learning benchmark with LLaMA-2 7B. Average accuracy (%) over all tasks is reported.

| mrr | trr | Avg. accuracy (%) |
|------|------|------|
| 0.10 | 0.80 | **79.6** |
| 0.05 | 0.40 | 51.5 |
| 0.05 | 0.70 | 75.9 |
| 0.20 | 0.90 | 71.1 |
| 0.50 | 0.50 | 55.8 |
| 0.70 | 1.00 | 48.0 |

**(2) Unconstrained Fine-Tuning of Low Singular Vectors:** In our method, gradient updates are projected back into the low-rank subspace to prevent interference with high-rank directions. This ablation removes that constraint: we freeze the high singular vectors but allow unconstrained updates to the low singular vectors, meaning that during optimization, updates are not restricted to stay within the initially identified low-rank subspace. This allows the low singular vectors to drift into the space previously occupied by high singular vectors, leading to potential interference and loss of previously acquired knowledge. As expected, this results in catastrophic forgetting, with accuracy dropping from 79.6% to 31.2%. In addition, since only the low singular vectors are updated while the high ones are frozen, each new task is forced to be learned in a restricted subspace, limiting the model's overall expressiveness. Together, these factors result in a $\approx 50$-point accuracy drop, highlighting the necessity of maintaining orthogonality between new task updates and previously learned subspaces.

Table 9: Ablation results on the LLaMA-2 7B model using the standard 5-task continual learning benchmark.

| Method | Average Accuracy (%) |
|------|------|
| OSFT (ours) | 79.6 |
| (1) Halved mrr/trr (aggressive effective rank approximation) | 51.5 |
| (2) No projection (unconstrained low-rank updates) | 31.2 |

## A.10 IMPLEMENTATION DETAILS

We detail the implementation of all experiments presented in this work. Our study utilizes both encoder-decoder and decoder-only language models. For all continual learning experiments—including the 5-task and 15-task benchmarks, as well as the TRACE benchmark—we replicate the task sequences, prompts, and dataset configurations as established in O-LoRA Wang et al. (2023a) and TRACE Wang et al. (2023b).

**T5-Large.** Experiments with the T5-Large model were conducted on a single NVIDIA H100 GPU using standard PyTorch training in full precision. We used a constant learning rate of $5 \times 10^{-5}$ with the AdamW optimizer and a total batch size of 8, training for one epoch per task. For each

classification dataset, we sampled 1,000 examples per class (where available) to construct balanced training sets, following the protocol established in Wang et al. (2023a). All runs were performed with a fixed random seed, and checkpoints were saved after each task for evaluation and reproducibility.

**LLaMA-2 7B and Mistral-7B.** All experiments with the LLaMA-2 7B and Mistral-7B models were conducted on a server equipped with 8 NVIDIA H100 GPUs, using the DeepSpeed library with Stage 2 optimization. Gradient checkpointing was enabled, and training was performed with a per-GPU batch size of 1 (resulting in an effective batch size of 8). We used the AdamW optimizer with a learning rate of $1 \times 10^{-5}$, weight decay of 0.01, $\beta_1 = 0.9$, $\beta_2 = 0.999$, and $\epsilon = 1 \times 10^{-8}$. All continual learning runs were trained for one epoch per task. After backpropagation, projection steps were applied to the gradients to constrain updates within the designated low-rank subspaces.

Our SVD configuration was automatically generated by analyzing specific matrices in each transformer block—namely, q_proj, k_proj, v_proj, o_proj, gate_proj, up_proj, and down_proj. Among the various strategies we explored for determining which singular vectors to retain, we found empirically that two approaches consistently performed best. The first allocates a fixed budget by freezing the top $\dfrac{i-1}{n}$ fraction of singular vectors for task $i$ in an $n$-task sequence. The second uses adaptive rank selection based on layer importance scores, as described in Section 3.4, where the number of retained singular vectors per layer is computed using the normalized importance $I^{(l)}$ from Section 3.3. The remaining components were fine-tuned using projected gradient descent within the low-rank subspace.

**Clarification on $mrr$, $trr$ choice and $I^{(l)}$ handling.** We introduced two key hyperparameters, minimum retention ratio (mrr) and target retention ratio (trr) in Eq. equation 3. Empirically, we set these values as $mrr = 0.1$ and $trr = 0.8$, which consistently yielded strong results across benchmarks with T5-Large, LLaMA-2 7B, and Mistral-7B. As shown in Appendix A.9 (Table 9), halving these parameters (more aggressive fine-tuning on the new task) significantly reduced performance (from 79.6% down to 51.5%), demonstrating the importance of appropriately balancing stability and plasticity. Conversely, mild perturbations around the defaults (e.g., ±0.02 or ±0.05) produced less than a 1% change in accuracy, confirming robustness. Thus, starting from our recommended default $(0.1, 0.8)$ and performing a modest grid search around these values on early tasks is a simple and effective strategy for practical tuning. Additionally, if prior knowledge about task difficulty is available, the retention rates can be adjusted accordingly.

Regarding negative layer importance scores $I^{(\ell)}$: By definition, we compute $I^{(l)}$ as the cosine similarity between the input activations $X^{(l)}$ and output activations $Y^{(l)}$. Empirically, we observed these values were consistently positive (ranging roughly between 0.5–0.8) across all layers and tasks. Nevertheless, for robustness, if any raw cosine similarity were negative, we explicitly clip it to zero before normalization, ensuring:

$$\sum_\ell I^{(l)} = L$$

This guarantees that the retention ratio

$$r^{(l)} \geq \mathrm{mrr} \times (\text{full rank}).$$

## A.11 RUNTIME AND RESOURCE ANALYSIS

To assess the deployment practicality of our method, we analyze both theoretical and empirical resource overheads compared to full fine-tuning (FFT).

**Memory Efficiency during Training.** The key efficiency gain comes during training. By freezing the high-rank components, we avoid storing their gradients and optimizer states. For an $n \times n$ matrix, the memory for parameters, gradients, and Adam optimizer states is roughly $4n^2$ for full fine-tuning, whereas our method requires approximately $2n^2 + 6nr$ (for storing U/V factors and low-rank optimizer states, where $r$ is the trainable rank). OSFT is therefore more memory-efficient whenever $r < n/3$, which typically holds in continual learning. In practice, $r < n/3$ is sufficient

because starting from an instruction-tuned model, most prior capabilities must be retained, and one-third of the subspace capacity is generally adequate to learn new tasks. This constant memory footprint is a significant advantage over methods that store past gradients or activation bases.

**SVD Time Complexity.** The time complexity of computing the SVD of an $n \times n$ weight matrix is $O(n^3)$. We perform SVD once per matrix at the start of each task. For LLaMA-2 7B, where typical weight matrices have dimensions $4096 \times 4096$ and the input sequence length is $L = 512$, this cost is on par with $\sim 4$ forward passes through a transformer block, since the cost of a single forward pass is $O(Ln^2 + L^2n)$. Empirically, performing SVD on all relevant matrices of LLaMA-2 7B takes approximately **2 minutes** on a single H100 GPU. This is less than 4% of the total training time for a typical 7k-sample dataset, becoming negligible for larger datasets.

**Empirical Runtime.** We profiled the wall-clock runtime of performing SVD on all relevant matrices in LLaMA-2 7B. Using a single H100 GPU:

- Time for SVD (all matrices): $\sim 2$ minutes
- Time for fine-tuning on a 7k-sample dataset for 3 epochs: $\sim 25$ minutes

Thus, the SVD overhead is $<10\%$ of total training time on small datasets, and becomes negligible on larger datasets. Moreover, the SVD cost scales linearly with the number of tasks and matrices, making it practical even in long-horizon continual learning.

**Rank Estimation Cost.** Rank selection based on capacity heuristics (e.g., dividing total budget by number of tasks) adds no computational cost. When using adaptive rank selection via input-output similarity, we only require a single forward pass over $\sim 500$ samples from the previous task—an overhead of under 30 seconds on modern GPUs.

**Comparison with Gradient Storage.** Many existing continual learning methods store task-specific gradients or importance weights. For each task, this requires storing an additional $n^2$ tensor per matrix, leading to $O(kn^2)$ total memory for $k$ tasks. Our method avoids this entirely, making it significantly more scalable.

Our method incurs only a small one-time per-task cost for SVD computation and requires no additional memory for gradient storage. Theoretical and empirical results confirm that it is both scalable and suitable for deployment in resource-constrained environments.

### A.12 TASK SEQUENCES AND PER-TASK PERFORMANCE

Across all three experimental settings—the 5-task standard CL benchmark, the 15-task longer sequence benchmark, and the 8-task TRACE benchmark—we strictly adhered to the original configurations of O-LoRA Wang et al. (2023a) and TRACE Wang et al. (2023b). This included using the same datasets, task instructions for prompting models during classification and generation, and identical training and validation sample counts and label distributions per task. Task sequences were replicated exactly to ensure consistency across evaluations and facilitate fair comparisons.

To better assess forgetting and learning dynamics, we report per-task performance for one representative task sequence from the 5-task (Table 11) and 15-task (Table 12) continual learning benchmarks, and the task sequence in the TRACE benchmark (Table 13). Each task reports two metrics: the accuracy immediately after the task is learned (reflecting plasticity), and the final accuracy after all tasks are trained (reflecting stability and forgetting).

These breakdowns provide a more granular view of both adaptation (learning) and forgetting across tasks. The final results in the main paper (e.g., Table 1) are averaged across multiple such task orders. The same orders were used for both T5-Large and LLaMA-2 7B experiments.

### A.13 CODE AVAILABILITY

The implementation of the OSFT training method, including dataset processing and training pipelines is available at:

Table 10: Task orders for all six sequences used in the Standard Continual Learning benchmark experiments.

| Order | Task Sequence |
|-------|---------------|
| 1 | dbpedia → amazon → yahoo → ag |
| 2 | dbpedia → amazon → ag → yahoo |
| 3 | yahoo → amazon → ag → dbpedia |
| 4 | mnli → cb → wic → copa → qqp → boolqa → rte → imdb 
 → yelp → amazon → sst−2 → dbpedia → ag → multirc → yahoo |
| 5 | multirc → boolqa → wic → mnli → cb → copa → qqp → rte → imdb 
 → sst−2 → dbpedia → ag → yelp → amazon → yahoo |
| 6 | yelp → amazon → mnli → cb → copa → qqp → rte → imdb 
 → sst−2 → dbpedia → ag → yahoo → multirc → boolqa → wic |

Table 11: Per-task performance on the 5-task benchmark (Order 1, T5-Large Model). Task order: dbpedia → amazon → yahoo → ag. Average Accuracy: 75.3, Backward Transfer: -3.7.

| Task | Accuracy (After Task) | Accuracy (Final) |
|------|----------------------|------------------|
| dbpedia | 98.9 | 97.2 |
| amazon | 53.5 | 46.1 |
| yahoo | 74.3 | 68.7 |
| ag | 89.1 | 89.1 |

https://github.com/Red-Hat-AI-Innovation-Team/mini_trainer

## A.14 USE OF LARGE LANGUAGE MODELS

Consistent with the ICLR 2026 disclosure policy, we used large language models (LLMs) only to aid or polish writing (grammar and English fluency) and for minor assistance in editing experimental code (e.g., debugging or syntax corrections). LLMs were not used for research ideation, theoretical analysis, methodology design, data analysis, or result generation. All scientific contributions, theoretical derivations, core code development, and experiments were implemented and validated solely by the authors.

Table 12: Per-task performance on the 15-task benchmark (Order 4, T5-Large Model). Task order: mnli → cb → wic → copa → qqp → boolqa → rte → imdb → yelp → amazon → sst-2 → dbpedia → ag → multirc → yahoo. Average Accuracy: 71.6, Backward Transfer: -5.5.

| Task | Accuracy (After Task) | Accuracy (Final) |
|---|---|---|
| mnli | 74.9 | 62.3 |
| cb | 83.6 | 75.0 |
| wic | 57.2 | 51.2 |
| copa | 54.4 | 47.0 |
| qqp | 85.8 | 82.4 |
| boolqa | 83.2 | 77.3 |
| rte | 83.4 | 80.5 |
| imdb | 96.1 | 94.5 |
| yelp | 59.7 | 48.6 |
| amazon | 54.2 | 49.1 |
| sst-2 | 92.3 | 93.3 |
| dbpedia | 98.5 | 94.6 |
| ag | 84.8 | 72.1 |
| multirc | 78.1 | 75.8 |
| yahoo | 69.8 | 69.8 |

Table 13: Per-task performance on the TRACE benchmark (LLaMA-2-7B-Chat). Task order: C-STANCE → FOMC → MeetingBank → Py150 → ScienceQA → NumGLUE-cm → NumGLUE-ds → 20Minuten. Average Accuracy: 48.4, Backward Transfer: -7.1.

| Task | Accuracy (After Task) | Accuracy (Final) |
|---|---|---|
| C-STANCE | 0.48 | 0.42 |
| FOMC | 0.65 | 0.58 |
| MeetingBank | 0.56 | 0.49 |
| Py150 | 0.60 | 0.48 |
| ScienceQA | 0.73 | 0.67 |
| NumGLUE-cm | 0.37 | 0.28 |
| NumGLUE-ds | 0.54 | 0.51 |
| 20Minuten | 0.44 | 0.44 |

