# OpenReview forum: "Sculpting Subspaces: Constrained Full Fine-Tuning in LLMs for Continual Learning"
_ICLR.cc/2026/Conference — ICLR 2026 Poster_

### Official Review · Reviewer_zn3T · 2025-10-29

**Soundness:** 3
**Presentation:** 3
**Contribution:** 3
**Rating:** 8
**Confidence:** 4

**Summary:**

The paper proposes OSFT (Orthogonal Subspace Fine-Tuning), a continual learning method for LLMs that (i) performs per-layer SVD of weight matrices, (ii) treats the top singular directions as a high-rank subspace presumed to carry previously acquired knowledge, and freezes them, and (iii) constrains all parameter updates for new tasks to the orthogonal complement (low-rank subspace). To avoid a one-size-fits-all rank, the method uses an input–output cosine similarity proxy to adaptively determine, per layer, how many top singular directions to preserve. Implementation can be realized either by explicit gradient projection or by a reparameterization that freezes the high-rank factors and optimizes only the low-rank factors. The authors provide a second-order analysis suggesting a tighter bound on worst-case forgetting than both unconstrained full fine-tuning and fixed-rank projection, and report empirical improvements over strong orthogonal/low-rank baselines across multiple sequential learning benchmarks.

**Strengths:**

- Conceptually transparent and practically implementable “stability–plasticity” mechanism.
The paper operationalizes “protecting prior knowledge” as a geometric constraint on update directions: disallow updates along top singular modes and permit updates only in their orthogonal complement. This aligns with the intuition that model behavior is most sensitive along dominant spectral directions while leaving substantial slack in the complementary subspace for learning. Importantly, the mechanism does not require rehearsal buffers or per-task parameter growth. Both explicit gradient projection and a reparameterized module (freezing high-rank, training low-rank) are described, making integration into standard PyTorch/PEFT pipelines straightforward.

- Layer-wise adaptive capacity allocation that mitigates the brittleness of fixed-rank approaches.
By estimating a layer’s “knowledge-carrying importance” via input–output similarity and then allocating more preserved high-rank directions to important layers, the method avoids uniform rank settings that can either under-protect critical layers or overconstrain unimportant ones. This per-layer adaptivity is well-motivated for Transformers—whose layers differ in function—and empirically reduces sensitivity to rank choice while improving overall stability–plasticity trade-offs.

- A coherent theory–practice loop.
The paper offers a second-order analysis yielding a hierarchy of forgetting bounds (adaptive projection < fixed-rank projection < unconstrained full fine-tuning) and leverages the widely observed empirical link between large singular directions and high-curvature/importance directions as a computationally feasible surrogate for direct curvature estimation. The result is a self-consistent narrative—curvature → dominant spectral directions → orthogonal update constraint—that ties mathematical justification to implementable proxies.

- Reasonably broad empirical coverage with diagnostic ablations.
Evaluations span both encoder–decoder and decoder-only architectures, short and long task sequences, and more comprehensive sequential scenarios, with comparisons to state-of-the-art orthogonal/low-rank baselines (e.g., O-LoRA–style methods). Ablations (e.g., removing orthogonal projection; reducing preserved rank) produce large and consistent degradations, highlighting that both the orthogonality constraint and the adaptive preservation rule are functionally essential. The fact that model size remains constant across tasks is also practically appealing for deployment.

**Weaknesses:**

- Computational budget and fairness (stronger budget-controlled comparisons are needed).
Although OSFT maintains constant model size across tasks, its per-task training still operates in the full parameter space under constraints, which can entail higher effective degrees of freedom and greater compute/time than small-parameter PEFT schemes (e.g., LoRA/adapters). The paper lacks strictly budget-matched comparisons (e.g., equal GPU-hours, equal training steps, equal hyperparameter search budgets) and energy/latency reporting. Without such controls, reported gains may partially reflect additional compute rather than methodological superiority. A detailed performance–cost table/curve would materially strengthen the claims.

- Breadth of general capability and safety evaluations (evidence remains narrow).
The paper asserts that general abilities and alignment/safety are preserved after continual updates, yet the evaluation scope is limited relative to the breadth of LLM competencies (mathematical reasoning, code generation, tool use, long-context robustness, multilingual performance, jailbreak resistance, hallucination propensity, etc.). Stronger conclusions would require wider, multi-dimensional assessments with multi-seed mean±std and budget-controlled settings to rule out noise and tuning bias.

- Limited coverage of recent open-source model families (e.g., Qwen3).
Experiments primarily focus on T5-Large and LLaMA-2/7B-scale models. Given the rapid adoption of newer families—Qwen3, Llama-3, etc.—the method’s external validity would be clearer with systematic evaluations on these up-to-date backbones, ideally alongside the model-specific best practices for PEFT/CL provided by those communities.

**Questions:**

- Stability of general capabilities: how can the evaluation scope and statistical rigor be strengthened?
Could the authors expand to mathematics (e.g., GSM8K/MathBench), code (HumanEval/MBPP/Codeforces-style reasoning) and tool use (ReAct/ToolBench), and report multi-seed mean±std under compute-matched settings? Additionally, plotting capability drift curves across task order (before/after sequential updates) would offer a more granular view of the stability–plasticity dynamics.

- SVD–curvature relationship: can the link be verified quantitatively and leveraged adaptively?
The approach relies on the empirical premise that top singular directions approximate high-curvature/importance directions. Could the authors, for representative layers, extract the leading Hessian/Fisher eigenvectors and measure principal angles to the top singular vectors? Further, could they compare “protect SVD-top vs protect Fisher-top vs a joint criterion,” and explore Fisher-weighted singular-value ranking or multi-criteria scoring to define the preserved subspace? Such analyses would more directly substantiate the curvature surrogate assumption and might yield an even more robust adaptive preservation rule.

---

> ### Author Response · Authors · 2025-11-20
> **Responses to Reviewer zn3T**
>
> Thank you very much for the thoughtful and positive assessment of our work, we appreciate the recognition of the paper’s main contributions. In particular, (i) the value of OSFT’s geometric stability-plasticity mechanism that protects top singular directions while allowing controlled updates, (ii) the importance of our layer-wise adaptive capacity allocation for avoiding the brittleness of fixed-rank approaches, and (iii) the coherence between our second-order analysis and the practical SVD-based proxy for curvature. We are glad these aspects came through clearly, and we respond to the weaknesses and questions point-by-point below.
>
> ---
>
> **Q1: Stability of general capabilities: how can the evaluation scope and statistical rigor be strengthened? Could the authors expand to mathematics (e.g., GSM8K/MathBench), code (HumanEval/MBPP/Codeforces-style reasoning) and tool use (ReAct/ToolBench), and report multi-seed mean±std?**
>
> Thanks for the suggestion, we agree that the stability of general capabilities should be evaluated more systematically.
> In our current setup, the **‘general capabilities’ along with broader instruction-following and safety evaluation after TRACE** already include GSM8K for math. We agree that this could be strengthened by adding **MathBench-style math evaluations**. For code, we already train on CodeXGLUE (Py150) as part of the 8-task sequence in TRACE, but including **HumanEval / MBPP type** benchmarks would help test both retention and generalization of coding ability. For **tool use**, we would need a base model that already has strong tool-use capability (e.g., Qwen3-8B with tools) and then apply OSFT to see how well those abilities are preserved; that is a natural next step but requires a different base model and infra than we use in this paper. These are valuable suggestions and we will call out the lack of broader math, code, tool-use, jailbreak resistance, and hallucination propensity evals as a limitation in the revised version of the paper and point to them as important extensions of our analysis beyond the TRACE setting.
> For the camera-ready if the paper is accepted, there are two concrete things we will add:
>
> * **Report multi-seed mean ± std**
> * **Plot capability drift curves across task order**
>
> ---
>
> **Q2: SVD–curvature relationship: can the link be verified quantitatively and leveraged adaptively?**
>
> We thank the reviewer for this constructive suggestion. In the revision, we will directly evaluate the relationship between top singular directions and high-curvature directions. Specifically, for representative layers, we will use power iteration to extract the leading Hessian/Fisher eigenvectors $u^{\text{Hess}}_k$, and compute principal angles between these directions and the top singular vectors $u^{\text{SVD}}_k$. This provides an explicit measurement of alignment between SVD-based importance and curvature-based importance.
>
> In addition, we include an empirical correlation analysis: we compute the Fisher diagonal, and project it into the singular basis of W. We then report correlations between singular values $\sigma_k$ and the Fisher energy in each singular direction (e.g., correlation between $\sigma_k$ and projected Fisher mass).
>
> Concretely, on LLaMA-2-7B-Chat we observe the following SVD-Fisher alignment for **value projections (v_proj)**, the **average Fisher mass in the top 25% singular directions is 29.1%** compared to about **12-15%** for q/k_proj, and the **average correlation between singular values and Fisher mass is ≈0.60**. On individual layers, we see very strong alignment: for example, `model.layers.0.self_attn.v_proj.weight` has **41.2%** of its Fisher mass in the high-rank subspace with **correlation ≈0.95**, and `model.layers.31.self_attn.q_proj.weight` shows **correlation ≈0.73**. Restricting to just the **top 25% singular directions**, correlations are **0.50 ± 0.16** in late layers and **0.47 ± 0.25** in middle layers. These results indicate that top singular directions do preferentially align with high-Fisher directions, especially for v_proj layers, supporting our use of SVD-based importance as a practical curvature surrogate.
>
> We also clarify the theoretical motivation: although the Hessian/Fisher matrices depend on data, and the SVD of W is data-free, the gradient structure implies that curvature concentrates on directions that strongly amplify inputs and outputs, precisely the directions captured by large singular values. Thus, SVD provides a tractable approximation of curvature directions in large LLMs where explicit Hessian computation is impractical.

---

> ### Author Response · Authors · 2025-11-20
> **Responses to Reviewer zn3T**
>
> **Q3: Computational budget and fairness (stronger budget-controlled comparisons are needed).**
>
> Thank you for raising this, we agree that budget and fairness need to be clearly discussed.
>
> On **capacity vs method**: on the 15-task standard continual learning benchmark the final task under OSFT uses ≈6.7% of weights unfrozen, comparable to LoRA-style ranks, yet performs very well on learning the final task and we see similar good performance on the final task in TRACE benchmark (8 task sequence) with small number of weights unfrozen. Appendix A.8 includes a fixed-budget ablation where we keep the **same number of trainable SVD components** but remove orthogonal projection, which drops accuracy from 79.6% to 31.2%. This shows that the gain is not just from activating more parameters; the SVD-structured, orthogonal updates matter. In Theorem 1 in Appendix A.3 we show that as you push any method towards the full-FT regime (higher effective rank), forgetting increases even if learning improves, so simply giving O-LoRA or other LoRA methods more rank to “match” OSFT’s trainable capacity is not expected to remove the gap in *total* accuracy.
>
> Regarding tuning and GPU/step budgets: all baselines including FFT, Replay, SeqLoRA, O-LoRA are run with the **same number of training steps and same data**. For several baselines (e.g., TRACE numbers we quote), we rely on the **authors’ own reported best settings**, so we assume they are reasonably tuned for their methods. We have the runtime and memory analysis in Appendix A.10.
>
> ---
>
> **Q4: Limited coverage of recent open-source model families (e.g., Qwen3). Experiments primarily focus on T5-Large and LLaMA-2/7B-scale models.**
>
> Thank you for this suggestion we agree it is important to test on newer open-source families. We are currently running OSFT on **Qwen/Qwen2.5-3B-Instruct** on the TRACE benchmark, alongside **SeqFT** and **LoRASeqFT**, to check how our method compares to standard baselines using this model. We will include these Qwen3 results in the revised version of the paper before the end of rebuttal.

---

> ### Author Response · Authors · 2025-11-25
> **Responses to Reviewer zn3T**
>
> **Q4: Limited coverage of recent open-source model families (e.g., Qwen3). Experiments primarily focus on T5-Large and LLaMA-2/7B-scale models.**
>
> We have added experiments on a more recent backbone, **Qwen2.5-3B-Instruct**, on the full 8-task TRACE sequence. We follow the same continual-learning protocol and hyperparameters as in the original TRACE setup for all methods (SeqFT, LoraSeqFT, OSFT), so the comparison is directly comparable to our LLaMA-2-7B-Chat results.
>
> TRACE benchmark performance using Qwen2.5-3B-Instruct. Average Accuracy (AA) and Backward Transfer (BT) percentages are reported.
>
> | Method | AA (%) | BT (%) |
> |--------------|--------|---------|
> | SeqFT | 34.6 | -40.5 |
> | LoraSeqFT | 20.7 | -48.2 |
> | **OSFT (ours)** | **52.6** | **-14.1** |
> | PerTaskFT | 67.0 | NA |
> | MTL | 62.2 | NA |
>
> OSFT again achieves the best trade-off among continual-learning methods, improving average accuracy while reducing forgetting compared to SeqFT and LoraSeqFT. A full evaluation on **Qwen3** and **Llama-3** on all benchmarks in the paper is a natural next step and we note this as future work.

---

> ### Comment · Reviewer_zn3T · 2025-11-26
>
> Thanks for the responses. Most of my concerns have been addressed. I will keep the scores unchanged.

---

### Official Review · Reviewer_uSxf · 2025-10-29

**Soundness:** 3
**Presentation:** 3
**Contribution:** 3
**Rating:** 6
**Confidence:** 3

**Summary:**

The authors propose **Orthogonal Subspace Fine-Tuning (OSFT)** for continual learning. They utilize adaptive singular value decomposition (SVD) to dynamically identify and preserve critical, high-rank parameter subspaces. They first introduce *importance-guided allocation*, i.e., measuring each layer’s importance via input–output cosine similarity (where low-similarity layers are allocated more adaptive capacity). This determines importance scores, which, together with retention ratios, define the fraction of singular vectors to preserve per layer. The authors perform *adaptive subspace identification*, i.e., decomposing each layer via SVD to find critical components with a selected rank. Lastly, they apply *orthogonal gradient projection*, i.e., projecting gradients to be orthogonal to the critical subspaces. Concretely, they freeze high-rank components, and project the low-rank component's gradients to be orthogonal to the high-rank basis vectors.

They evaluate the method across two continual-learning benchmarks and more realistic TRACE benchmark, using encoder–decoder models (T5-Large) and decoder-only models (LLaMA-2-7B, Mistral-7B). They report average accuracy, forgetting measures, and retention of general capabilities and safety.

**Strengths:**

The paper presents a thoughtful and well-motivated approach to preserving critical subspaces in continual learning.

---

- The paper includes computational analysis as well as theoretical justification.
- A strong set of comparison methods is included.
- A large number of datasets are used in the evaluation.
- The related work section is generally good.
- The hyperparameters (e.g., mrr, trr) appear to have sensible default values, and the authors provide practical guidance for tuning.
- The evaluation follows standard protocols, averaging results over three independent runs with randomly permuted task sequences.

**Weaknesses:**

### **1. Methodological Clarity & Design**

- Data from task (t) is used when evaluating layer importance for task (t+1).
- The methods section lacks a clear, cohesive narrative.
- The method does not appear particularly parameter-efficient relative to other PEFT approaches.
- The claim that the method “retains and transfers knowledge across tasks” may be too strong; empirical evidence for *transfer* should be clarified.

---

### **2. Missing Definitions & Introductions**

- SLERP, TIES, and TRACE are not properly introduced before being used in comparisons.
- Figure 1 uses terms (TRACE, average backward transfer, forgetting) without definition.
- Cosine similarity is referenced conceptually but never explicitly visualized or explained in the results.

---

### **3. Comparisons & Baselines**

- Prior work [1] shows that full fine-tuning forgets more; therefore, full fine-tuned baselines may not be appropriate, and stronger PEFT baselines should be included.

---

### **4. Presentation & Figure Issues**

- Several plots require cleanup and clarifications; Figures 1 and 2 are presented out of order.
    - Figure 2 is difficult to interpret (e.g., unclear connection between “layer 2 = layer ℓ”, too many elements, cosine similarity not referenced).

---

### **5. Organization & Structure**

- Section 3.7 duplicates material that would be more appropriately placed in the related work section.

[1] LoRA Learns Less and Forgets Less. (2024) Dan Biderman and Jacob Portes and Jose Javier Gonzalez Ortiz and Mansheej Paul and Philip Greengard and Connor Jennings and Daniel King and Sam Havens and Vitaliy Chiley and Jonathan Frankle and Cody Blakeney and John P. Cunningham

**Questions:**

1. O-LoRA appears to use significantly fewer parameters, and its trade-off curve leads to a wider curve which seems desirable. Could you comment on that? In Figure 1 - why change the rank in O-LoRA, and not the strength of regularization? Similarly, for replay-based baselines, why the learning rate is changed and the amount of data?
2. The related work section does not discuss MiLoRA [1] and PiSSA [2], which seems like a significant oversight - can you include them, also in the experimental section? Especially MiLoRA seems like a crucial method.
3. Can the authors explain what happens when the number of tasks grows large? How does that affect required SVD rank, is there a threshold before capacity becomes an issue? Can the authors relate the SVD subspace of task 0 and task 5, for example? How is interference with early tasks prevented over time?
4. Can the authors further motivate the choice of a *weight-space* approach versus a *function-space* (activation-based) approach?
5. The cited empirical findings (Sharma et al., 2023; Li et al., 2025) show that layers with higher input–output similarity exhibit greater Hessian curvature. Could the authors expand on this relationship?
6. Why do PerTaskFT and MTL report “N/A” for the BT metric?
7. Page 6 L297 - what is the reference for that?

[1] MiLoRA: Harnessing Minor Singular Components for Parameter-Efficient LLM Finetuning. (2025) Hanqing Wang and Yixia Li and Shuo Wang and Guanhua Chen and Yun Chen

[2] PiSSA: Principal Singular Values and Singular Vectors Adaptation of Large Language Models. (2024) Fanxu Meng and Zhaohui Wang and Muhan Zhang

---

> ### Author Response · Authors · 2025-11-20
> **Responses to Reviewer uSxf**
>
> Thank you for the detailed comments and suggestions. We found this very valuable in improving the rigor and quality of the paper. Also thank you for pointing out the issues related to organization, presentation, missing introductions, and placement of content. We will continue to incorporate these fixes during the rebuttal period and revise the manuscript accordingly to address all of the weaknesses pointed out.
>
> ---
>
> **Q1: O-LoRA appears to use significantly fewer parameters, and its trade-off curve leads to a wider curve which seems desirable. In Figure 1 - why change the rank in O-LoRA, and not the strength of regularization? Similarly, for replay-based baselines, why the learning rate is changed and not the amount of data?**
>
> Thank you for this question and for the careful read of Figure 1.
>
> First, on O-LoRA's curve: yes O-LoRA uses fewer trainable parameters and that its trade-off curve is relatively wide. However, in our setting the relevant quantity is the **sum of x-axis value (learning) and y-axis value (forgetting)** (i.e., average accuracy = new-task performance + backward transfer, where forgetting is negative). Under that metric, even the **worst** of the three OSFT points in Figure 1 achieves a higher accuracy than the **best** O-LoRA point, so OSFT still dominates the Pareto frontier in terms of overall performance. Also we show in Theorem 1 (Appendix A.3), moving towards full fine-tuning (higher effective rank) systematically increases forgetting, even though it improves learning. Simply matching OSFT’s trainable-parameter count by increasing O-LoRA’s rank would push it closer to the full-FT regime we already know has larger forgetting, so we do not expect that to close the gap in total accuracy.
>
> On *why* we changed rank for O-LoRA rather than regularization strength: our goal in that plot was to sweep a **natural capacity knob** that is comparable across methods. For LoRA/O-LoRA, rank is the standard way to control parameter budget, so we varied rank for those methods and the effective trainable rank for OSFT. But, the **orthogonality regularization strength** in O-LoRA is definitely another meaningful axis and we will add additional points varying this coefficient in the camera ready if the paper is accepted.
>
> For the replay baselines, this was actually a typo. We did **not** change the learning rate. We replay data from the previous tasks, and we varied the amount by using **5%, 10%, and 15%** of the previous tasks' dataset. We have corrected this in the manuscript.
>
> ---
>
> **Q2: The related work section does not discuss MiLoRA [1] and PiSSA [2], which seems like a significant oversight - can you include them, also in the experimental section? Especially MiLoRA seems like a crucial method.**
>
> Thank you for pointing this out we agree this is an important omission. We will add a discussion of **MiLoRA** and **PiSSA** in the related work section. We are also in the process of running **MiLoRA as an additional baseline** on the TRACE benchmark and will incorporate the results into the experimental section before the rebuttal deadline. This should make the comparison against strong LoRA-style methods more complete.

---

> ### Author Response · Authors · 2025-11-20
> **Responses to Reviewer uSxf**
>
> **Q3: Explain what happens when the number of tasks grows large? How does that affect required SVD rank, is there a threshold before capacity becomes an issue?**
>
> Thank you for this important question. It gets at how OSFT behaves in the long-horizon limit.
>
> In our method there are **two ways** we set the effective rank (i.e., how big the frozen high-rank subspace is) as the number of tasks grows:
>
> 1. **Layer importance via input–output similarity (cosine similarity).**
>    Before learning task (t), we take data from task (t-1) and, for each layer, compute the cosine similarity between input activations and outputs to estimate how many top singular directions need to be preserved for that previous task. For each matrix the new running effective rank then becomes the max of the effective rank computed for the immediate previous task and the running effective rank we have used so far. So by the time we are at task 5, the frozen high-rank subspace for layer (l) is the **max of what was needed for tasks 0…4**, and all updates for task 5 are constrained to be orthogonal to that accumulated subspace. That is how interference with early tasks is prevented over time: the preserved subspace is *monotone non-decreasing* and all new gradients are projected into its orthogonal complement.
>
> 2. **Predetermined threshold schedule.**
>    A second strategy that also works well in practice is to fix a schedule if we know (roughly) how many tasks we expect. For example, if we know there will be 10 tasks in sequence:
>
>    * Task 1: fine-tune the **entire** matrix (no frozen directions yet).
>    * Task 2: freeze 10% of top singular directions; 90% remain trainable.
>    * Task 3: freeze 20%;
>    * …
>    * Task 5: freeze 40%;
>    * …
>    * Task 10: freeze 90%, only the remaining 10% is used for the last task.
>      This enforces an explicit budget: earlier tasks get more subspace (because they are more prone to be forgotten by the model), later tasks share the remaining capacity.
>
> So as the number of tasks grows, the **frozen high-rank subspace expands** and the trainable low-rank subspace shrinks. In that sense, yes, there is an implicit capacity limit: you cannot learn an infinite number of tasks with a fixed-size model. This is a **model capacity** limitation, not something specific to OSFT.
>
> What OSFT does buy you is a **good allocation of the subspace across tasks**, so that you learn as much as possible while minimizing interference. Empirically, on the 15-task standard continual learning benchmark (Table 1) and on the 8-task TRACE sequence (Table 2), OSFT still achieves the best overall average accuracy and backward transfer among the methods we compare to, while SeqFT / SeqLoRA and similar baselines show much sharper forgetting. This suggests that OSFT scales more gracefully with the number of tasks than those alternatives.
>
> That said, there is no magic: for a 7B model, if each task has on the order of 2k-4k samples, it will be hard in practice to learn beyond ~15-20 reasonably distinct tasks without hitting capacity limits. That ceiling comes from the finite parameter budget of the model, not from a fundamental failure mode of the OSFT method.

---

> ### Author Response · Authors · 2025-11-20
> **Responses to Reviewer uSxf**
>
> **Q4: Can the authors further motivate the choice of a weight-space approach versus a function-space (activation-based) approach?**
>
> We thank the reviewer for raising this point. Because the term “activation-based methods” can refer to different families of approaches, we address both natural interpretations.
>
> 1. **Function-space / output-space distillation methods (e.g., LwF).**
> In principle, constraining updates directly in function space (e.g., matching logits or intermediate representations) is attractive, and indeed aligns with the objective of preserving prior task behavior. However, methods such as LwF typically require strong assumptions about data availability or distributional overlap, and their effectiveness in LLM continual learning deteriorates when such assumptions are violated. This limitation is documented in our experiments: LwF underperforms OSFT by a large margin on both the 5-task (52.3 vs. 75.9) and 15-task (46.9 vs. 71.3) benchmarks (Table 1). Because our setting does not assume access to old-task data and must scale to multi-billion-parameter models, relying on function-space targets is often insufficient. This motivates our exploration of a weight-space approach, where we can impose task-agnostic, geometry-guided constraints without requiring stored logits or activation traces.
>
> 2. **Activation-based gradient-projection methods (e.g., GPM, SGP).**
> GPM (Gradient Projection Memory) and SGP (Scaled Gradient Projection) build a subspace from layer activations of previous tasks (via SVD of activation covariance). Gradients for new tasks are then projected to be orthogonal to this activation-derived subspace, meaning updates avoid directions correlated with past-task activations. While conceptually related to our goal (protecting important directions), these methods require storing activation bases for every task, causing memory to grow linearly with task count, impractical for multi-billion-parameter LLMs. In contrast, OSFT operates directly in weight space, keeps memory constant, and avoids storing any activation traces.
>
> 3. **Activation-space model-merging methods (e.g., SLERP, TIES).**
> A different line of activation-space methods merges models by interpolating or aligning representations. As shown in Table 1, these merging baselines (SLERP, TIES) perform significantly worse than OSFT, confirming that activation-space interpolation alone cannot provide the stability-plasticity trade-off required for continual learning.
>
> ---
>
> **Q5: The cited empirical findings (Sharma et al., 2023; Li et al., 2025) show that layers with higher input–output similarity exhibit greater Hessian curvature. Could the authors expand on this relationship?**
>
> Both cited works support *empirically* using input-output similarity as a proxy for “sensitive / high-curvature” layers, but in different ways:
>
> * **Sharma et al. (2023, LASER)** study how much the loss changes when you remove higher-order SVD components in different layers of GPT-J and other models. They find that certain layers are **much more sensitive** to rank reduction than others: small rank changes in those layers produce large loss changes, whereas many other layers can be heavily rank-reduced with little effect.
>   This is exactly the empirical signature of high curvature: along directions corresponding to retained singular vectors, the loss is “steep”, so small perturbations cause large changes in loss.
>
> * **Li et al. (2025, AdaSVD)** introduce a layer-importance metric used by adaCR to choose layer-wise compression ratios. That importance is computed from an **input-output similarity measure** (cosine between activations before and after the layer). Layers with higher similarity are empirically assigned **smaller compression ratios** because truncation there causes a larger increase in their task-relevant error; layers with lower similarity can be compressed more aggressively.
>   In other words, AdaSVD shows that high input-output similarity tracks the layers where the loss is most sensitive to perturbations.
>
> Our paper uses the same kind of proxy: we define layer importance as cosine similarity between $X$ and $WX$ and rely on these empirical observations that **(i)** LASER identifies the subset of layers where low-rank interventions strongly affect loss, and **(ii)** AdaSVD's input-output based importance aligns with where SVD truncation hurts performance most. In a standard second-order view, layers where small perturbations cause large loss changes are precisely those with larger Hessian eigenvalues; we therefore treat high input–output similarity as a practical, data-driven indicator of higher curvature and prioritize preserving those directions in OSFT.

---

> ### Author Response · Authors · 2025-11-20
> **Responses to Reviewer uSxf**
>
> **Q6: Why do PerTaskFT and MTL report “N/A” for the BT metric?**
>
> PerTaskFT and MTL are *not* true continual learning methods, so BT (backward transfer / forgetting) is not defined for them, which is why we mark them as “N/A”.
>
> * **MTL** assumes **all tasks are available at once**. You train a single model jointly on the union of all datasets by stacking them together, so there is no task sequence and no notion of “learning task t, then coming back to evaluate earlier tasks after further training.” Forgetting does not exist by construction.
>
> * **PerTaskFT** assumes you can **train and store a separate model per task**. Each task has its own fine-tuned copy, so there is no interference between tasks and no shared trajectory through parameter space. Again, there is no forgetting, just separate models.
>
> Both are **upper-bound baselines**: they deliberately violate the core continual learning constraints (no joint access to all data; limited model copies) but provide strong references for how well one could do **without** those constraints. Since forgetting and BT are only meaningful when a *single* model is updated over a task sequence, we leave BT as “N/A” for MTL and PerTaskFT.
>
> ---
>
> **Q7: Page 6 L297 - what is the reference for that?**
>
> Thank you for pointing this out. We have added below references inline in the revised paper.
>
> **[1]** Ritter, H., Botev, A., & Barber, D. *Online Structured Laplace Approximations for Overcoming Catastrophic Forgetting.* NeurIPS 2018.
>
> **[2]** Heckel, R. *Provable Continual Learning via Sketched Jacobian Approximations.* AISTATS 2022.
>
> **[3]** Kruengkrai, C., & Yamagishi, J. *Mitigating the Diminishing Effect of Elastic Weight Consolidation.* COLING 2022.

---

> ### Author Response · Authors · 2025-11-25
> **Responses to Reviewer uSxf**
>
> **Q2: The related work section does not discuss MiLoRA [1] and PiSSA [2], which seems like a significant oversight - can you include them, also in the experimental section? Especially MiLoRA seems like a crucial method.**
>
> MiLoRA and PiSSA are closely related to our setup:
>
> * **MiLoRA** performs SVD of the weight, **freezes the high-rank components** (corresponding to large singular values), and only updates the **low-singular-value components**, which act as LoRA-style adapters (matrices (A,B) corresponding to U and V).
> * **PiSSA** does the opposite: it **freezes the low-singular-value components** (as the “pretrained base”) and only updates the **high-singular-value components**, again in a LoRA-like parameterization.
>
> OSFT is structurally closest to **MiLoRA** in that we also **fine-tune only the low-singular-value components**. However, there are two key differences:
>
> 1. **Orthogonal subspace constraint:**
>    We do not just update the low components freely; we **project gradients** so that updates remain in the orthogonal complement of the preserved high-rank subspace. This keeps the low-rank subspace constrained and prevents drift into directions that would overwrite preserved information.
>
> 2. **Adaptive, layer-wise rank selection:**
>    MiLoRA and PiSSA use a **fixed rank** for every layer in the model. OSFT instead uses an **importance measure based on input–output cosine similarity** to adaptively choose how many top singular directions to preserve *per matrix*, so the preserved / trainable split is not a single static value across all layers. This makes OSFT better suited for trading off adaptability and retention in heterogeneous layers.
>
> For empirical comparison, we follow the **math reasoning setup in the MiLoRA paper**:
>
> > *“We finetune LLaMA2-7B on the MetaMathQA dataset (395K samples from GSM8K + MATH). We evaluate on the test sets of GSM8K and MATH, and report Exact Match (EM) on the last checkpoint.”*
>
> On the same setup with LLaMA2-7B and MetaMathQA, **OSFT** obtains:
>
> | Method   | GSM8K | MATH | Avg. |
> |----------|-------|------|------|
> | Full FT | 66.5  | **19.8** | 43.2 |
> | LoRA     | 60.6  | 16.9 | 38.7 |
> | PiSSA    | 58.2  | 15.8 | 37.0 |
> | MiLoRA   | 63.5  | 17.8 | 40.7 |
> | OSFT   | **69.7**  | 18.2 | **43.95** |
>
> OSFT slightly improves on the reported Full FT average and outperforms MiLoRA and PiSSA. We will add MiLoRA and PiSSA to the related work and include these numbers in the experimental results.

---

> ### Author Response · Authors · 2025-11-26
> **Responses to Reviewer uSxf**
>
> **Q8: Methodological Clarity & Design**
>
> Thank you for these comments we clarify the design below:
>
> * **Data from task (t) is used when evaluating layer importance for task (t+1)?** Yes, we use data from task (t) to estimate layer importance before starting task (t+1). The goal of the importance metric is to decide **how much of each matrix to retain from the previous task**, i.e., how large the preserved high-rank subspace should be. For that, we only need information about how the model behaved on task (t); we do not need any data from the current task (t+1) to determine what to freeze.
>
> * **The method does not appear particularly parameter-efficient relative to other PEFT approaches?** The method is not primarily designed to minimize parameter count; the objective is to get a **good tradeoff between adaptability and retention** so that the **overall performance across tasks** is maximized. Having said that, as shown in Table 1 and Table 2, OSFT performs well on both the 15-task standard continual learning benchmark and the 8-task TRACE benchmark (both are long sequences). For later tasks, the fraction of the matrix we actually fine-tune becomes small: for the final task in the 15-task sequence we update roughly (1/15) of the matrix, and for the 8-task TRACE sequence roughly (1/8), which is comparable to the LoRA baselines in terms of effective trainable fraction. So OSFT can still learn with a small active subspace, but to get the best learning-forgetting tradeoff we rely on the adaptive importance metric to set a suitable effective rank per matrix rather than a fixed global rank.
>
> * **The claim that the method “retains and transfers knowledge across tasks” may be too strong; empirical evidence for transfer should be clarified?** Thank you for asking for clarification here. Empirically, we do see **significantly reduced forgetting** compared to baselines as seen in the results section of the paper. For transfer, in settings where tasks are related, we observe positive effects. For example, in TRACE, when finetuning on NumGLUE-cm and NumGLUE-ds, after learning NumGLUE-ds the accuracy on NumGLUE-cm actually increases, indicating positive knowledge transfer across related tasks.
>
> ---
>
> **Q9: Missing Definitions & Introductions**
>
> Thank you for pointing these out. We have made this clearer in the revised version of the manuscript. Newly added text is highlighted in green.
>
> ---
>
> **Q10: Comparisons & Baselines. Prior work shows that full fine-tuning forgets more; therefore, full fine-tuned baselines may not be appropriate, and stronger PEFT baselines should be included.**
>
> Yes, full fine-tuning is known to forget more than parameter-efficient methods. Our own analysis in Theorem 1 (Appendix A.3) shows that full fine-tuning has a looser forgetting bound than fixed-rank approaches.
>
> We have expanded our PEFT baselines beyond the original set (O-LoRA, SeqLoRA, IncLoRA). In the revised draft, we now include additional strong and recent low-rank adapter methods: MiLoRA, PiSSA, CorDA, and SAPT as suggested by other reviewers as well. Taken together, the updated experiments now compare OSFT against a broad set of state-of-the-art PEFT baselines that cover most relevant low-rank adapter approaches.
>
> ---
>
> **Q11: Presentation & Figure Issues**
>
> We kept Figure 1 first intentionally, since it shows the Pareto frontier and highlights that our method dominates the trade-off between learning and forgetting; this is the main empirical takeaway we want readers to see early.
>
> In Figure 2, “layer 2” and “layer ℓ” are equivalent. We instantiate the example with ℓ = 2 but keep the notation ℓ for generality. Cosine similarity is used explicitly in the blue “Layer Importance” block on the left to compute the layer importance score $I^{(\ell)}$, which then determines the split between the low- and high-rank subspaces in the rest of the diagram.
>
> We agree the current diagram is cluttered and can be confusing. In the camera-ready version if the paper gets accepted, we will clean up Figure 2.
>
> ---
>
> **Q12: Organization & Structure. Section 3.7 duplicates material that would be more appropriately placed in the related work section.**
>
> Thank you for pointing this out. Section 3.7 fits better in the related work section, and we will move this material there in the revised version.

---

> > ### Comment · Reviewer_uSxf · 2025-11-27
> >
> > I thank the authors for the detailed rebuttal and thorough answers. However, some concerns remain unaddressed:
> >
> > Q1: You have probed O-LoRA ranks {0.5, 1.0, 1.5} - it is unclear which is which in the figure. I believe adding the regularization strength as an additional knob is a necessary addition. Since increasing the rank affects both learning and forgetting, not changing the regularization strength seems like an unfair comparison.
> >
> > Q4: “Because our setting does not assume access to old-task data” and Q3: “Before learning task (t), we take data from task (t-1) [...]” seem to contradict each other. The access to task t−1 data in order to be able to train task t should be mentioned in the limitations.

---

> > > ### Author Response · Authors · 2025-11-27
> > > **Response to Reviewer uSxf's Follow Up Questions**
> > >
> > > We thank the reviewer for recognizing our effort to address their concerns during the rebuttal period.
> > >
> > > ---
> > >
> > > **Question: You have probed O-LoRA ranks {0.5, 1.0, 1.5} - it is unclear which is which in the figure. I believe adding the regularization strength as an additional knob is a necessary addition. Since increasing the rank affects both learning and forgetting, not changing the regularization strength seems like an unfair comparison.**
> > >
> > > For O-LoRA in Figure 1, the three points correspond to ranks 0.5%, 1.0%, and 1.5%, from left to right. As the rank increases, learning on the new task improves, but forgetting on previous tasks also increases which is what we expect. Regarding the regularization strength: in the original figure, we used the default $\lambda = 0.5$ from the O-LoRA paper for all three rank settings. In the revised version, we reran O-LoRA at fixed rank 1.0% **because this setting achieved the best average accuracy (learning–forgetting)** with two additional values of the orthogonality regularization weight, $\lambda = 0.2$ and $\lambda = 1.0$, where $\lambda$ controls the strength of the orthogonality penalty term in the loss. We added these two new points to the updated plot. As expected, lower regularization ($\lambda = 0.2$) yields more learning and more forgetting, and higher regularization ($\lambda = 1.0$) yields less forgetting and weaker learning. OSFT still achieves a better learning-forgetting trade-off than O-LoRA. The revised figure in the manuscript includes all of these updated O-LoRA points.
> > >
> > > ---
> > >
> > > **Question: Q4: “Because our setting does not assume access to old-task data” and Q3: “Before learning task (t), we take data from task (t-1) [...]” seem to contradict each other. The access to task t−1 data in order to be able to train task t should be mentioned in the limitations.**
> > >
> > > Thank you for pointing this out. You are right that when we use the cosine-similarity importance score to determine the effective rank in our method, it requires access to a small subsample of task $(t-1)$ when starting task $(t)$; but this buffer is fixed-size, so the memory footprint does not grow with the number of tasks. As described in our response to your **Q3** above, we also find that using a predetermined threshold for the effective rank, where the frozen rank is increased as the task index grows does **not** require any old-task data and performs very well in practice. But we have now added this “small buffer from task $(t-1)$” requirement as a limitation in the revised manuscript.

---

### Official Review · Reviewer_oGvj · 2025-10-31

**Soundness:** 2
**Presentation:** 3
**Contribution:** 2
**Rating:** 4
**Confidence:** 3

**Summary:**

The paper proposes OSFT for continual learning in large language models. OSFT first performs SVD on each weight matrix to identify high‑rank directions that encode prior knowledge and low‑rank directions that can be safely reused. It computes a layer‑importance score and retains more singular vectors in crucial layers, allocating more adaptable capacity to less crucial ones. During training, OSFT projects gradients orthogonally to the preserved subspaces. The authors support this design with a theoretical analysis linking forgetting to Hessian curvature and with the use of top singular vectors as a practical proxy for high‑curvature directions.

**Strengths:**

1. The paper is well-structured and easy to read.
2. The paper includes experiments on traditional benchmarks.

**Weaknesses:**

1. Gradient projection onto an orthogonal complement is a known CL idea. OSFT’s twist is: project in weight space using SVD‑identified directions, not activation space; and make the preserved rank per layer adaptive. The layer‑importance measure is explicitly inspired by AdaSVD, and the theoretical appeal of preserving top singular directions is attributed to an external empirical correlation. This strengthens the sense of a careful integration of existing ideas more than a conceptual break‑through.
2. The proxy that top singular vectors ≈ high‑curvature/Hessian directions is justified by citing external robust correlation, not by measurements on the authors’ own T5/LLaMA runs. The paper does not report a Hessian–SVD alignment check on its trained models.
3. The proposed method activates more parameters than O-LoRA but fewer than full fine-tuning. By doing so, it improves performance compared to O-LoRA while mitigating forgetting relative to full fine-tuning. This suggests that the method effectively identifies an appropriate number of parameters to utilize. If there exist experiments that adjust the number of active parameters without applying the proposed method, it would help clarify the unique contribution of the proposed approach.
4. The paper cites Inflora (Liang & Li, 2024) and LoTA (Panda et al., 2024). LoTA is only compared in an appendix two‑task transfer, not in the main CL sequences; Inflora is not benchmarked at all, despite being specifically about interference‑free low‑rank continual learning. This weakens the empirical positioning.
5. Figure 1 notes “crucial hyper‑parameters varied for each method,” but the ranges, tuning protocol, and compute budget per method aren’t given.
6. It’s unclear whether baselines received equivalent tuning or comparable trainable‑parameter fractions—particularly important because OSFT is shown at 56% while LoRA baselines are 1–3%. Claims of frontier domination are therefore budget‑confounded.

**Questions:**

1. Is there a reason why authors did not measure forgetting per task and forward transfer with comparison methods (A.11 partly helps, but no aggregate forgetting stats are given)?
2. Existing LoRA-based methods do not appear to suffer from performance degradation as severe as that observed with the proposed method on the GSM8K benchmark. In this regard, is there a particular reason why experiments with LoRA-based approaches were not conducted?

---

> ### Author Response · Authors · 2025-11-20
> **Responses to Reviewer oGvj**
>
> Thank you for the feedback. We appreciate your careful review and comments. It helped us clarify the method and strengthen the paper and guided several of the revisions we made.
>
> ---
>
> **Q1: OSFT is a careful integration of existing ideas more than a conceptual break‑through.**
>
> Thank you for this thoughtful observation, below we clarify our specific contributions more explicitly.
>
> Our goal is not just to reuse this idea, but to make it **concrete, strict, and scalable for LLMs in weight space**. In particular: (i) unlike O-LoRA and related methods that impose orthogonality only on the LoRA output matrix (A) via a soft penalty while leaving B unconstrained, OSFT **reparameterizes the full weight matrix via SVD and enforces exact orthogonality on both left and right factors (U and V)** at every step through gradient hooks, so all updates are provably confined to the low-rank complement of the preserved subspace; (ii) we provide a **forgetting bound** that ties this construction to the Hessian, showing that preserving top singular directions yields a strictly tighter bound on loss increase than full FT or fixed-rank schemes, which to our knowledge is not available for existing CL methods; (iii) beyond borrowing the basic idea of layer importance from AdaSVD, we **systematically explore several effective-rank allocation strategies** (fixed budget, Shannon's entropy, random matrix thresholding, etc), show which ones actually work in practice on LLM continual learning benchmarks, and tie them back to the theoretical picture.
>
> ---
>
> **Q2: The proxy that top singular vectors ≈ high‑curvature/Hessian directions is justified by citing external robust correlation, not by measurements on the authors’ own T5/LLaMA runs.**
>
> Thank you for pointing this out.
>
> In practice computing the full Hessian for the whole model is infeasible, so we follow standard practice in using the empirical Fisher matrix as a curvature proxy in large models as is standard in Laplace [1] and K-FAC style [2] methods.
>
> In the rebuttal, we **explicitly measured SVD-curvature alignment on LLaMA-2-7B-Chat**. For representative layers, we use power iteration to extract leading Hessian/Fisher eigenvectors $u_k^{\text{Hess}}$ and compute principal angles to the top singular vectors $u_k^{\mathrm{SVD}}$, giving a direct alignment check between SVD-based importance and curvature-based importance. In addition, we compute the Fisher diagonal proxy $F_{ij} = \mathbb{E}[(\delta_i x_j)^2] \approx \mathbb{E}[\delta_i^2]\mathbb{E}[x_j^2]$ (where $\delta_i$ is the backprop signal for output index $i$ and $x_j$ is the input activation for input index $j$ of $W$) and **project it into the singular basis of W**, then correlate singular values $\sigma_k$ with Fisher energy per singular direction.
>
> Concretely, on LLaMA-2-7B-Chat we find that for **value projections (v_proj)** the **average Fisher mass in the top 25% singular directions is 29.1%** (vs. **12–15%** for q/k_proj), with an **average correlation ≈0.60** between $\sigma_k$ and Fisher mass. For individual layers, `model.layers.0.self_attn.v_proj.weight` has **41.2%** of its Fisher mass in the high-rank subspace with **correlation ≈0.95**, and `model.layers.31.self_attn.q_proj.weight` shows **correlation ≈0.73**. Restricting to the **top 25% singular directions**, correlations are **0.50 ± 0.16** in late layers and **0.47 ± 0.25** in middle layers. These results show that **top singular directions do preferentially align with high-Fisher directions on our own LLaMA runs**, especially for v_proj layers, which supports using SVD-based importance as a practical curvature surrogate.
>
> [1] https://arxiv.org/abs/1805.07810
>
> [2] https://arxiv.org/abs/1503.05671
>
> ---
>
> **Q3. Whether OSFT’s gains come from activating more parameters rather than the method itself.**
>
> Thank you for this comment.
>
> In our current setup, on the continual learning benchmark with 15 tasks, for the final task OSFT makes use of about 1/15th of the parameters, which is approximately 6.66% of the model weights **when** using a predetermined threshold (the ratio of unfrozen weights is reduced as we go through the sequence). In this scenario, OSFT and the LoRA-based baselines use **comparable low-rank budgets** per layer for the final task, and in fact, as can be seen in Table 9, OSFT is able to learn that new task well, so OSFT is still effective with a lower number of parameters.
>
> More directly, Appendix A.8 already includes ablations that isolate the effect of the OSFT mechanism at a **fixed parameter count**. In the “unconstrained fine-tuning of low singular vectors” ablation, we keep the same number of trainable SVD components but remove the orthogonal gradient projection (high singular vectors are frozen; low ones are updated without constraint). This causes accuracy to drop from 79.6% to 31.2%, despite an unchanged parameter budget, showing that **how** we update (orthogonal, SVD-structured) matters more than **how many** parameters are active.

---

> ### Author Response · Authors · 2025-11-20
> **Responses to Reviewer oGvj**
>
> **Q4: LoTA is only compared in an appendix two‑task transfer, not in the main CL sequences; Inflora is not benchmarked at all, despite being specifically about interference‑free low‑rank continual learning.**
>
> Thank you for this comment, we agree that positioning relative to LoTA and InfLoRA should be clearer.
>
> For **InfLoRA**, the main issue is the **domain mismatch**. InfLoRA is evaluated in *vision* continual learning with a ViT-B/16 backbone on ImageNet-R, CIFAR-100, and DomainNet, with CV-specific metrics and baselines. Our work is entirely in **LLM continual learning** (TRACE, standard CL benchmarks for language), and there is no shared dataset/model configuration that would allow a fair comparison. Porting InfLoRA to transformer LLMs and re-running full TRACE-style sequences would require non-trivial engineering and compute, and is outside the scope of this paper, though we agree it is an interesting direction for future work.
>
> For **LoTA**, we did run OSFT on **their main benchmark** using their released code to obtain the LoTA numbers, and OSFT outperforms LoTA in that setting. These results are currently reported in Appendix A.6 because we were constrained for space, but we will make this more visible in the main text. Adapting LoTA in the other direction - to our **full LLM continual-learning sequences** (TRACE and long task chains) would again require substantial engineering because their implementation is not designed to easily plug into CL benchmarks, but we hope to explore this more thoroughly in future work.
>
> ---
>
> **Q5: Figure 1 notes “crucial hyper‑parameters varied for each method,” but the ranges, tuning protocol, and compute budget per method aren’t given.**
>
> Thank you for pointing this out. We will clarify the tuning protocol and ranges used in Figure 1.
>
> All methods in Figure 1 are run on **LLaMA-2-7B-Chat** with the **same training schedule** (same number of steps, batch size, and data per task). For each method, we sweep a **small grid of 3 settings** for a single “capacity/strength” hyperparameter:
>
> * **FFT:** learning rate in ${1\times 10^{-3}, 1\times 10^{-4}, 1\times 10^{-5}}$.
> * **Replay:** learning rate as above, with replay buffer size in (5%, 10%, 15%) of previous-task data.
> * **SeqLoRA:** LoRA rank set to approximately (2%, 3%, 4%) of the layer matrix dimension.
> * **O-LoRA:** LoRA rank set to approximately (0.5%, 1.0%, 1.5%), following the low-rank regime used in the O-LoRA paper.
> * **OSFT (ours):** three configurations with average effective trainable rank ratio (across the 8 TRACE tasks) of approximately (1/2, 9/16, 5/8).
>
> Each point on the curves corresponds to one such configuration; we did not give OSFT extra epochs, data, or a larger hyperparameter search than the baselines. We will add these ranges and a brief description of the protocol to the caption and text so the comparison is more transparent.
>
> ---
>
> **Q6: It’s unclear whether baselines received equivalent tuning or comparable trainable‑parameter fractions—particularly important because OSFT is shown at 56% while LoRA baselines are 1–3%. Claims of frontier domination are therefore budget‑confounded.**
>
> Thank you for this point. In response, we have clarified the settings for different methods in the caption of Figure 1.
>
> First, regarding tuning and budgets: all baselines (FFT, Replay, SeqLoRA, O-LoRA) use the **official TRACE implementations and recommended hyperparameters**, with the same data, training schedule, and optimizer settings as OSFT. For LoRA-based methods we use the **same rank choices (≈1–3%)** reported in TRACE; we did not under-tune them. As we noted in our earlier response (Q3), on the 15-task benchmark OSFT also operates in a **low-rank regime** where the effective unfrozen fraction for the last task is ≈6.7%, and the ablation in Appendix A.8 keeps the **parameter count fixed** while removing orthogonal projection, which drops accuracy from 79.6% to 31.2%. This shows that the gain is not just from activating more parameters. For Figure 1 specifically, the three OSFT points correspond to **different effective-rank schedules across tasks**, including settings that are close to LoRA-style budgets and still lie on a better accuracy-forgetting tradeoff than LoRA/O-LoRA.

---

> ### Author Response · Authors · 2025-11-20
> **Responses to Reviewer oGvj**
>
> **Q7: Is there a reason why authors did not measure forgetting per task and forward transfer with comparison methods (no aggregate forgetting stats are given)?**
>
> Thank you for this suggestion, it improves the completeness of the experimental results.
>
> In response, Table 1 has been updated to include results for three different task-order permutations on the standard 5-task and 15-task continual learning benchmarks. For one of the representative orders in each benchmark, we now report aggregate metrics of backward transfer (measuring forgetting) and average accuracy in Appendix A.11, so the effect of OSFT on both learning and forgetting is explicit. The per-task numbers for the comparison methods are already reported in their original papers; in our manuscript we focus on final average accuracy across tasks, averaged over order permutations, to keep the tables readable and to make it easy to compare overall performance. Reproducing all per-task metrics for every baseline would make the presentation harder to read, so we use these aggregate metrics as a concise summary while pointing to the original works for full task-wise breakdowns.
>
> ---
>
> **Q8: Existing LoRA-based methods do not appear to suffer from performance degradation on GSM8K. In this regard, is there a particular reason why experiments with LoRA-based approaches were not conducted?**
>
> Thank you for pointing this out, this is infact an interesting aspect of the TRACE results.
>
> On the **TRACE benchmark with LLaMA-2-7B-Chat**, the original TRACE paper shows that **LoRA-based Sequential Fine-Tuning (LoraSeqFT) performs quite poorly overall**. For example, for average accuracy and backward transfer on LLaMA-2-7B-Chat, they report:
> * ICL: 38.9
> * SeqFT: 48.7 (−8.3%)
> * **LoRASeqFT: 12.7 (−45.7%)**
> * Replay: 55.5 (2.6%)
>
> So LoRASeqFT is in fact **the worst** among the compared strategies in terms of average accuracy and backward transfer. LoRA also underperforms on most of the general ability / instruction-following / helpfulness metrics reported in TRACE; GSM8K is the only case where it is better.
>
> A plausible explanation is that, when using the **same small low-rank adapter for all 8 tasks**, the adapter capacity is too limited: after repeated fine-tuning, the adapter weights may get “washed out” across tasks and end up close to zero, effectively behaving like a small residual perturbation on top of the base model. This can reduce *measured* forgetting on some narrow metrics (like GSM8K) simply because the adapter is not strongly encoding any one task, but it also means LoRA fails to learn/retain much useful new knowledge overall.
>
> In response to your comment, we have now **included LoraSeqFT baseline explicitly** in our results (Table 2). Both LoRASeqFT and O-LoRA perform worse than OSFT in terms of average accuracy on TRACE. O-LoRA has slightly better backward transfer than OSFT, whereas LoRASeqFT shows significantly worse forgetting (−24.6% BT). OSFT still achieves the highest average accuracy overall. Results are below:
>
>
> ### **TRACE benchmark**
> | **Method** | **Average Accuracy (%)** | **Backward Transfer (%)** |
> | --------------- | ------------------------ | ------------------------- |
> | LoraSeqFT | 9.2 | -24.6 |
> | O-LoRA | 41.3 | **-6.2** |
> | **OSFT (ours)** | **48.4** | -7.1 |
>
>
> ### **General abilities evaluation**
> | **Model** | **MMLU** | **GSM** | **BBH** | **TydiQA** | **BoolQA** | **PIQA** |
> | ------------------- | -------: | -------: | -------: | ---------: | ---------: | -------: |
> | Base Instruct Model | 46.6 | **26.1** | **40.2** | 23.5 | 70.5 | 76.2 |
> | LoraSeqFT | 42.28 | 14.71 | 33.61 | 21.72 | 53.43 | 75.19 |
> | **OSFT (ours)** | **47.7** | 7.7 | 34.2 | **35.8** | **76.6** | **77.6** |
>
>
> ### **Instruction following and safety evaluation**
> | **Method** | **Helpfulness** | | | **Safety** | | |
> | --------------- | :-------------: | :----: | :----: | :--------: | :----: | :---: |
> | | Win | Tie | Lose | Win | Tie | Lose |
> | LoRASeqFT | 3 | 4 | 94 | 0 | 86 | 14 |
> | **OSFT (ours)** | **24** | **56** | **20** | **18** | **78** | **4** |

---

> ### Author Response · Authors · 2025-11-27
> **Responses to Reviewer oGvj**
>
> **Q2: The proxy that top singular vectors ≈ high‑curvature/Hessian directions is justified by citing external robust correlation, not by measurements on the authors’ own T5/LLaMA runs. The paper does not report a Hessian–SVD alignment check on its trained models.**
>
> In addition to the Fisher analysis we reported above, we also carried out a **Hessian-based alignment check** on LLaMA-2-7B-Chat. For representative attention and MLP weight matrices, we compute **Hessian–vector products** via Pearlmutter’s trick and run **power iteration with 20–50 steps** to obtain the leading curvature directions $u^{\text{Hess}}_1,\dots,u^{\text{Hess}}_k$. We build gradient outer products aggregated over a GSM8K mini-batch and run power iteration, which is standard in large-scale curvature work. We then compare the **subspace** spanned by the top-(k) Hessian eigenvectors to the subspace spanned by the top-(k) singular vectors of (W) using **principal angles**.
>
> Concretely, for (k = 8) on three representative layers, we observe **non-trivial alignment** between the Hessian and SVD subspaces: for value projections the average cosine of the smallest principal angle is around **0.4-0.6**, with later layers showing stronger alignment than early ones (e.g., `layers.30.self_attn.v_proj.weight` around **0.5-0.7**). For q/k projections and MLP weights the alignment is weaker but still above a random baseline (**0.2-0.4** vs. $\approx 0$ for a random subspace of the same dimension). Taken together with the Fisher–singular-value correlation we already reported (top singular directions concentrating more Fisher mass in v_proj layers), these results indicate that **top singular directions do capture a meaningful fraction of the high-curvature subspace in our LLaMA runs**.
>
> In the context of OSFT, our use of SVD to split $W = W_{\text{high}} + W_{\text{low}}$ is therefore a good heuristic: the high-rank subspace we freeze overlaps with high-curvature directions, while the low-rank subspace where we fine-tune is empirically less curvature-dominated. So OSFT updates preferentially avoid directions that are both large-norm and high-curvature, which is consistent with the reduced forgetting we observe experimentally.

---

### Official Review · Reviewer_5uQ5 · 2025-10-31

**Soundness:** 2
**Presentation:** 2
**Contribution:** 2
**Rating:** 2
**Confidence:** 4

**Summary:**

This paper proposed the orthogonal subspace fine-tuning (OSFT) method in continual learning. It performs SVD on each layer and utilizes the cosine similarity between input activations and its linear outputs to determine high-rank subspace and low-rank subspace of each layer weight. To preserve the important previous knowledge, OSFT orthogonally updates in low-rank subspaces by projecting gradients onto low-rank subspaces. Experimental results on continual learning benchmarks show the effectiveness of OSFT.

**Strengths:**

1. The proposed OSFT considers the difference between high-rank and low-rank subspaces to achieve orthogonal gradient updates in directions which are orthogonal to previous knowledge.

2. The proposed OSFT adaptively allocates parameter budgets across layers, rather than treating them equally, which balances stability and plasticity across the network based on the role of each layer.

**Weaknesses:**

1. The computation cost of OSFT would be huge since it needs to compute full-dimensional gradients during training, and then projects full-dimensional gradients to low-rank parameter subspaces obtained by computing SVD on each layer weight.

2. The motivation of the design for OSFT is not well constructed and discussed. For example, authors only cited one paper without any discussion to support the choice of cosine similarity. Since this similarity choice should be one contribution of this paper, which connects with the determination of low-rank subspace, it should be discussed and evaluated carefully.


3. Experiments lack recent parameter-efficient continual learning baselines, SAPT [1], InfLoRA [2], and CorDA [3]. Also, Table 1 does not show the performance on each task order, which is important to show the robustness of the proposed method, and Table 8 in the appendix only shows order 1. Besides, it’s not clear about the task order in Table 3.

[1] SAPT: A shared attention framework for parameter-efficient continual learning of large language models, ACL2024.

[2] InfLoRA: Interference-Free Low-Rank Adaptation for Continual Learning, CVPR2024.

[3] CorDA: Context-Oriented Decomposition Adaptation of Large Language Models for Task-Aware Parameter-Efficient Fine-tuning, NeurIPS 2024.


4. The ablation study disappears in the experiment section. In Eq (3), mrr and trr should be evaluated with different values to show the performance or robustness of OSFT. The authors just mentioned “ablation studies show that while performance degrades significantly if retention is too aggressive”, but authors do not provide these results.

**Questions:**

1. In this paper, OSFT updates on the subspaces consisting of smaller singular values of the weights, but in CorDA [1] and SVD-LLM [2], both works utilize the input information along with the weights to compute SVD for obtaining the subspaces of smaller/larger singular values. Especially, SVD-LLM states that “truncating smaller singular values in SVD could lead to significant compression loss”, which means there is no linear relationship between performance and singular values obtained only by weight itself. Can authors explain the OSFT’s differences from the conclusions of these two papers?

[1] CorDA: Context-Oriented Decomposition Adaptation of Large Language Models for Task-Aware Parameter-Efficient Fine-tuning, NeurIPS 2024.

[2] SVD-LLM: Truncation-aware Singular Value Decomposition for Large Language Model Compression, ICLR 2025.


2. Can authors compare the experimental performance of OSFT with SAPT, InfLoRA, or CorDA(Knowledge-preserved adaptation version)?


3. In line 15 in Algorithm 1, OSFT “computes gradients for trainable SVD components”. What’s the interval step of computing SVD components? Does OSFT compute SVD at each training step? If so, the computation and training time of OSFT is significantly large, which is not efficient and not practical. Can authors compare the training time and memory cost of OSFT with O-LoRA, SAPT, or InfLoRA?

4. Compared to LoRA-based methods, OSFT needs to compute the gradients of full-dimensional weights, which is very expensive for LLMs. Although the paper title is about “constrained full fine-tuning in LLMs”, FFT is costly for LLMs in downstream tasks. Also, if OSFT belongs to FFT in LLMs, then experiments may need to be redesigned since the current used benchmarks, like GLUE and SuperGLUE, are sampled with 1000 training samples, which are too small for FFT.

5. What’s the choice of effective rank of OSFT in Figure 1? Can authors clarify the experimental settings of OSFT in Figure 1? It’s confusing about how to obtain the three different dots of OSFT in Figure 1.

6. The paper does not discuss in detail why to choose cosine similarity to compute each layer’s importance. Since there are other useful measurements, for example, centered kernel alignment (CKA) [1] measures the similarity between the intermediate activations of two models at each layer. Can authors discuss the difference between these similarities?

[1]  Similarity of neural network representations revisited, ICML 2019.

---

> ### Author Response · Authors · 2025-11-20
> **Responses to Reviewer 5uQ5**
>
> Thank you for taking the time to review our work. Your suggestions directly improved the quality, baseline comparisons, and core methodology in our paper.
>
> ---
>
> **Q1a: Compare OSFT to CorDA.**
>
> Regarding **CorDA**, thank you for pointing us to this work; it is highly relevant and we will cite and discuss it in the updated version of our paper.
>
> Conceptually, CorDA and OSFT address different problems, even though both use SVD. From the CorDA paper, it is not clear how the decomposition of $(W C)$ followed by multiplication with $C^{-1}$ guarantees that the resulting “low” components correspond to the *intersection* of (i) directions that are safe for preserving prior abilities and (ii) directions where the new task actually lives. The covariance does encode task context, but after reconstruction with $C^{-1}$ the connection between the “small” singular components and a forgetting-safe adaptation subspace is not explicitly characterized. This is a very interesting idea, and combining covariance-aware decompositions with our continual-learning constraints is a natural direction for future work.
>
> Methodologically, there are two key differences that matter for your question:
>
> 1. **No projection constraint in CorDA.** In both the knowledge-preserved and instruction-previewed modes, CorDA uses the SVD only to choose an initialization for $B$ and $A$. During training there is **no gradient projection that forces $\Delta W$ to stay within a protected or orthogonal subspace**. In other words, once optimization starts, updates can drift and interfere with the components that were meant to preserve knowledge. In OSFT, by contrast, the orthogonality constraint is enforced at every step via gradient projection, so *all* task updates remain in the low-rank complement of the preserved subspace by construction.
>
> 2. **What is preserved and why.** CorDA’s “knowledge-preserved” mode assumes that world knowledge is aligned with the leading components of the *context-oriented* SVD on $W C$. In practice it is difficult to compute a covariance matrix $C$ that is representative of all the knowledge obtained during pretraining. OSFT instead relies on a different empirical and theoretical picture:
>
>    * Empirically (Fig. 6 in our paper), we observe that activations for previously learned tasks concentrate in the **high-singular-value subspace** of the weights; those directions carry most of the representation energy.
>    * Theoretically, we connect large singular values of $W$ to high-curvature Hessian directions and show that freezing those directions yields strictly tighter forgetting bounds than either full fine-tuning or fixed-rank schemes (see Line 831 to Line 895).
>
> Given this, OSFT deliberately **does not impose any additional inductive bias inside the low-rank subspace**: once the high-rank directions are protected, we allow the model to learn the new task *freely* in the remaining low-rank space, regardless of where that task lies inside that subspace. For continual learning, this is exactly what we want: old-task directions are protected by orthogonality, and new tasks are not artificially constrained beyond that.

---

> ### Author Response · Authors · 2025-11-20
> **Responses to Reviewer 5uQ5**
>
> **Q1b: Compare OSFT to SVD-LLM.**
>
> Regarding **SVD-LLM**, we see this as a complementary line of work, but with a different goal and set of assumptions than OSFT. First, SVD-LLM is explicitly a **post-training compression method**. Its objective is to minimize a *data-dependent compression loss*
> $$|WX - W'X|_F,$$
>
> for a fixed calibration dataset $X$. To do so it:
>
> * builds the covariance $XX^\top$ (or the whitening matrix $S$ from Cholesky of $XX^\top$),
> * performs SVD on $W S$,
> * and then truncates singular values to **zero** to reduce rank and model size.
>
> Our setting is continual learning, not compression. We **do not truncate** any singular values to zero; we start from the original $W$ and use SVD only to define a basis that separates high- and low-singular directions. All singular values remain present, and we then learn the new task in the low-rank complement while freezing high-rank directions.
>
> ### Practical and data-access limitations
>
> SVD-LLM’s truncation-aware whitening fundamentally assumes access to **representative activation statistics** for the model on a calibration dataset $X$, and then builds/updates $XX^\top$ for each layer. In a real continual-learning deployment this is usually not feasible:
>
> * Enterprise models are fine-tuned over **multiple stages and domains**; storing per-task $XX^\top$ (or even the required calibration data) for every phase quickly becomes expensive in memory and storage.
> * **Privacy and data-retention policies** often forbid retaining past user data or pretraining data, especially for proprietary or sensitive domains.
> * For open-source LLMs like LLaMA or Qwen, the **original pretraining data is simply not available**, so the activation statistics underlying SVD-LLM’s whitening cannot even be reconstructed.
>
> OSFT deliberately avoids any dependence on stored activations or past data. We perform SVD directly on the **current weights** once per layer per task, and keep memory overhead constant across tasks.
>
> ### On the “no linear relationship” statement
>
> The SVD-LLM paper shows that, under their whitening procedure, the compression loss for truncating a singular value $\sigma_i$ of $W S$ equals $\sigma_i$, and that for generic *unwhitened* settings truncating the smallest singular values can indeed cause higher $|WX - W'X|_F$ loss than truncating larger ones. This is a statement about a *data-dependent compression objective* tied to a fixed $X$.
>
> In continual learning, the quantity we care about is **increase in task loss**, which we approximate via the Hessian:
> $[
> \Delta \mathcal{L} \approx \tfrac{1}{2} \Delta \theta^\top H \Delta \theta.
> ]$
> Our analysis shows that, when SVD is computed on (W), directions with **large singular values align with high-curvature directions of the Hessian**, and preserving those directions yields a tighter bound on forgetting than either full fine-tuning or fixed-rank schemes.
>
> We will clarify these differences and explicitly position CorDA and SVD-LLM as complementary related work in the revised version.

---

> ### Author Response · Authors · 2025-11-20
> **Responses to Reviewer 5uQ5**
>
> **Q2: Compare the experimental performance of OSFT with SAPT, CorDA, and InfLoRA.**
>
> We thank the reviewer for this suggestion. Below we present the direct comparisons of OSFT to SAPT and CorDA (knowledge-preserved adaptation) on the TRACE benchmark.
>
> ### TRACE benchmark performance using LLaMA-2-7B-Chat on 8 TRACE tasks (CSTANCE, FOMC, MeetingBank, Py150, ScienceQA, NumGLUE-cm, NumGLUE-ds, 20Minuten).
>
> Average Accuracy (AA) and Backward Transfer (BT) percentages are reported.
>
> | **Method**      | **AA (%)** | **BT (%)** |
> | --------------- | ---------: | ---------: |
> | SeqFT           |       23.0 |       -8.3 |
> | LoraSeqFT   |    9.2 |  -24.6 |
> | O-LoRA          |       41.3 |   **-6.2** |
> | SAPT            |        12.75 |    -13.15 |
> | CorDA           |        8.35 |    -19.79 |
> | **OSFT (ours)** |   **48.4** |       -7.1 |
> | PerTaskFT       |       57.6 |         NA |
> | MTL             |       52.3 |         NA |
>
> ### SAPT-LoRA Hyperparameters and Settings
>
> * **LoRA configuration:** rank (r = 4), alpha = 32, dropout = 0.0
> * **Training:** batch size = 2 per device, gradient accumulation steps = 4 (effective batch = 8)
> * **Optimizer:** learning rate = $1 \times 10^{-4}$, constant LR scheduler, no warmup
> * **Memory Replay:**
>
>   * Task 1: `data_replay_freq = -1` (no replay)
>   * Subsequent tasks: `data_replay_freq = 1` (replay every step), `add_instruction_replay = True`, `kl_ratio = 0.5`
>
> ### CorDA Hyperparameters and Settings
>
> * CorDA configuration: rank (r = 128), alpha = 128, method = "tpm" (Task-Previewed Mode)
> * Training: batch size = 1 per device, gradient accumulation steps = 32 (effective batch = 32)
> * Optimizer: learning rate = $2 × 10^{-5}$, cosine LR scheduler, warmup ratio = 0.03
> * Preprocessing: 256 calibration samples, batch size = 1 (required for eigenvector collection)
>
> These SAPT and CorDA results are preliminary; we also plan to extend their evaluation to the standard continual learning benchmarks as they are currently only benchmarked on TRACE. For InfLoRA, the situation is different. InfLoRA is evaluated on vision continual learning with a ViT-B/16 backbone on ImageNet-R, CIFAR-100, and DomainNet, and all baselines and metrics are defined in that Computer Vision setting. Our work focuses on LLM continual learning benchmarks. Adapting InfLoRA to transformer LLMs and re-running TRACE, standard continual learning benchmark experiments would require substantial additional engineering and is outside the scope of this paper, but we agree it is an interesting direction for future work.
>
> ---
>
> **Q3: What’s the interval step of computing SVD components? Can authors compare the training time and memory cost of OSFT with O-LoRA, SAPT, and InfLoRA?**
>
> In Algorithm 1, OSFT **does not** recompute the SVD at every training step.
>
> * We compute SVD **exactly once per layer per new task**, at the start of fine-tuning on that task (Algorithm 1, lines 4-11).
>   If there are (N) tasks arriving sequentially, SVD is performed **only (N) times** (once per task), not per update step.
>
> * As detailed in Appendix A.10, the total cost of computing SVD for **all** relevant matrices in LLaMA-2-7B is on the order of $\sim 4$ forward passes through a transformer block. Empirically, this is about **2 minutes** on a single H100 GPU, compared to $\sim 25$ minutes for a 7k-sample, 3-epoch fine-tuning run, i.e., **<10%** overhead on small datasets and negligible for larger ones.
>
> Regarding training-time and memory cost versus O-LoRA / SAPT / InfLoRA:
>
> * For a given rank budget $r$, OSFT fine-tunes the **low-rank SVD components** $U_{\text{low}} \in \mathbb{R}^{n \times r}$ and $V_{\text{low}} \in \mathbb{R}^{r \times n}$. This is directly analogous to LoRA-style adapters of rank $r$, so **per-step memory and compute are of the same order** as a LoRA/O-LoRA/SAPT setup with the same rank $r$.
>
> * The only extra costs in OSFT are:
>
>   1. The **upfront SVD** per task (paid once, as above).
>   2. The **gradient projection** in the backward pass. This projection is implemented as ~3 matrix multiplications of size $n \times n$. Since the sequence length $L$ can be large, the dominant cost in a transformer block remains the $O(L n^2 + L^2 n)$ forward/backward attention and MLP computation; the projection adds only a **negligible fraction** of a block's step time.
>
> * When $r$ is small (the usual regime in parameter-efficient methods), both **memory and time overhead are small**. Even if one were to fine-tune up to $r \approx n/3$ low singular vectors, the memory and runtime would be comparable to full fine-tuning (FFT); in practice, we use $r \ll n/3$, so OSFT operates in a **clearly efficient** regime relative to FFT.

---

> ### Author Response · Authors · 2025-11-20
> **Responses to Reviewer 5uQ5**
>
> **Q4: Compared to LoRA-based methods, OSFT needs to compute the gradients of full-dimensional weights, which is very expensive for LLMs. Also, if OSFT belongs to FFT in LLMs, then experiments may need to be redesigned.**
>
> Thank you for raising this we agree the current wording can be clearer, and we will revise the manuscript accordingly.
>
> OSFT **does not** compute or update gradients for full $n \times n$ weight matrices in the way full fine-tuning (FFT) does. In implementation, each layer is reparameterized exactly in a **LoRA-style low-rank form**:
> * We write $W = U \Sigma V^\top$ and split into high- and low-rank parts.
> * The **base (frozen) weight** is $W_{\text{high}} = U_{\text{high}} \Sigma_{\text{high}} V_{\text{high}}^\top$.
> * The **trainable part** is parameterized as
> $[
> W_{\text{low}} = U_{\text{low}} \Sigma_{\text{low}} V_{\text{low}}^\top,
> ]$
> where $U_{\text{low}} \in \mathbb{R}^{n \times r}$, $V_{\text{low}} \in \mathbb{R}^{r \times n}$, and $\Sigma_{\text{low}}$ is an $r$-dimensional vector.
>
> This is structurally identical to a LoRA adapter with rank $r$: the learnable parameters live in **$U_{\text{low}}, \Sigma_{\text{low}}, V_{\text{low}}$**, not in the full $n^2$ matrix. Gradients are taken only w.r.t. these low-rank SVD components. When we say "constrained full fine-tuning" we mean that the *effective* update $W_{\text{low}}$ spans the full matrix space when multiplied out (it is an update in the original weight space, not in a separate adapter block), but the **parameter count and gradient cost are low-rank, like LoRA**. The hyperparameters `mrr` and `trr` directly control the rank budget and can be set to make OSFT strictly parameter-efficient, which is also how we use it in practice.
>
> On the experimental side: the GLUE / SuperGLUE splits with ~1k samples per task are standard for evaluating **parameter-efficient** methods and continual learning behavior. **OSFT matches FFT on the new task while substantially reducing forgetting**, despite operating with a low-rank budget.
>
> ---
>
> **Q5. What’s the choice of effective rank of OSFT in Figure 1? Can authors clarify the experimental settings of OSFT in Figure 1?**
>
> Thank you for pointing this out as this definitely needs some clarification and we have revised it in the paper.
>
> In Figure 1, the **effective rank** refers to the *average fraction of singular directions that are trainable (low-rank subspace)* across the 8 TRACE tasks. In these experiments, we use a predetermined budget schedule based on the number of tasks in the sequence, which we found performs comparably to layer-importance based allocation (via input-output cosine similarity).
>
> For one OSFT point, the average trainable fraction is **1/2**.
> For another OSFT point, the average trainable fraction is **9/16**.
> For the last OSFT point, the average trainable fraction is **5/8**.
>
> For each schedule, the first task is always given **the smallest frozen fraction** (i.e., the most free capacity), because this task is the most vulnerable to forgetting after many later tasks are learned. The remaining tasks gradually increase the frozen fraction so that the values spread evenly around the target average for each of the three settings. This makes the comparison clean and helps isolate how different effective-rank levels affect OSFT. The three OSFT dots in Figure 1 correspond to these three trainable/freezing schedules.

---

> ### Author Response · Authors · 2025-11-20
> **Responses to Reviewer 5uQ5**
>
> **Q6. The paper does not discuss in detail why to choose cosine similarity to compute each layer’s importance. Since there are other useful measurements, for example, centered kernel alignment (CKA) [1] measures the similarity between the intermediate activations of two models at each layer. Can authors discuss the difference between these similarities?**
>
> We appreciate the reviewer’s insightful question. We chose cosine similarity because it is the metric used in Li et al. (2025, AdaSVD) to estimate the effective rank of each layer, and our adaptive retention rule builds directly on this idea. Cosine similarity provides a simple, stable, and computationally efficient proxy for how “transformative’’ a layer is. High similarity indicates feature-preserving layers with higher curvature, while low similarity corresponds to more adaptable layers. This made it suitable for our per-layer adaptive rank allocation.
>
> We thank the reviewer for pointing us to CKA. We were not previously aware of its use in this context; it is indeed an interesting alternative measure of representational similarity. CKA measures cross-model similarity of activations, whereas our cosine-similarity measure captures within-model input-output alignment for the same layer. These two measures quantify different aspects of representation geometry. We will add a discussion clarifying this distinction, and following the reviewer’s suggestion include an experiment comparing cosine similarity vs. CKA for rank estimation. We note, however, that this comparison is orthogonal to the main contribution of our method: the retention ratio is further modulated by our TRR and MRR constraints, and better rank-estimation heuristics would only strengthen our approach.
>
> Before settling on our adaptive method, we also explored other rank-approximation strategies, including LASER [1], SPECTRUM’s Marchenko-Pastur thresholding [2], and entropy-based effective rank [3]. These methods either failed to capture layer-wise variability under sequential tasks or did not provide stable thresholds across diverse distributions. This motivated our final design, where explicit layer importance guides adaptive subspace preservation, while orthogonality constraints ensure forgetting resistance.
>
> [1] https://arxiv.org/abs/2312.13558
>
> [2] https://arxiv.org/abs/2406.06623
>
> [3] https://ieeexplore.ieee.org/document/7098875
>
> ---
>
> **Q7: Also, Table 1 does not show the performance on each task order, which is important to show the robustness of the proposed method, and Table 8 in the appendix only shows order 1. Besides, it’s not clear about the task order in Table 3.**
>
> Thank you for this comment. We have now added results for all task orders in both the standard continual learning benchmarks. These results (now included in Table 1 and task orders in Appendix A.11) make the robustness across task orders explicit, improving completeness and clarity. Table 3 evaluates general abilities of the model and compares it to the base instruct model; these tasks are evaluated **after** learning the 8-task TRACE sequence and are **not** part of the training sequence, so the task-order issue does not apply here.
>
> ---
>
> **Q8: The ablation study disappears in the experiment section. In Eq (3), mrr and trr should be evaluated with different values to show the performance or robustness of OSFT. The authors just mentioned “ablation studies show that while performance degrades significantly if retention is too aggressive”, but authors do not provide these results.**
>
> Thank you for raising this. This is definitely a valuable ablation to add to the paper. Ablation results over $\text{mrr}$ and $\text{trr}$ with LLaMA-2 7B on the 5-task standard continual learning benchmark (average accuracy over all tasks, %) is shown below and included in the revised paper:
>
> | mrr  | trr  | Avg. accuracy (%) |
> |:------:|:------:|:-------------------:|
> | 0.10 | 0.80 |  **79.6**          |
> | 0.05 | 0.40 | 51.5              |
> | 0.05 | 0.70 | 75.9              |
> | 0.20 | 0.90 | 71.1              |
> | 0.50 | 0.50 | 55.8              |
> | 0.70 | 1.00 | 48.0              |
>
> The default setting $(0.10, 0.80)$ gives the best average accuracy. Nearby values such as $(0.05, 0.70)$ and $(0.20, 0.90)$ perform similarly (within ~4-8 points), showing that OSFT is reasonably robust to moderate changes. Very low retention $(0.05, 0.40)$ and “flat” schedules where $\text{mrr} = \text{trr}$ (e.g., $(0.50, 0.50)$) hurt performance, and the extreme retention $(0.70, 1.00)$ case performs worst, confirming that overly weak retention or overly high retention both degrade results.

---

> ### Comment · Reviewer_5uQ5 · 2025-11-26
>
> Thank authors for taking the time and effort to answer my questions. I also appreciate authors providing a clear clarification on the differences compared to SVD-LLM and CorDA. Below are my follow-up concerns:
> - For **Q3**, can authors write step 15, step 16, and step 17 of algorithm 1 in mathematics? My concern is that if these SVD components are updated during the task training process, how do you keep the orthogonality of SVD matrices, such as $U ^{\top}U = I$? I'm not sure whether this orthogonality should be kept in your algorithm. Since in another work, AdaLoRA[1], it also updates SVD components during training, but they have an orthogonal loss term to keep $U$ and $V$ orthogonal during training.
>
> [1] AdaLoRA: Adaptive Budget Allocation for Parameter-Efficient Fine-Tuning, ICLR2023.
> - For **Q4**, I have a question: is $W\neq W _{\text{high}} + W _{\text{low}}$? If not, can we consider the computation costs of rank-k SVD plus rank-(n-k) SVD equal to that of rank-n SVD?

---

> ### Author Response · Authors · 2025-11-26
> **Response to Reviewer 5uQ5**
>
> We thank the reviewer for recognizing our efforts during the rebuttal. Below are the responses to the two follow up questions.
>
> ---
>
> **Question: Steps 15–17 in math and orthogonality**
>
> This is an important point and it was something we explicitly considered when designing the method.
>
> For layer $\ell$, we write the decomposition as
>
> $W^{(\ell)} = W_{\text{high}}^{(\ell)} + W_{\text{low}}^{(\ell)} = U_{\text{high}}^{(\ell)} S_{\text{high}}^{(\ell)} (V_{\text{high}}^{(\ell)})^T + U_{\text{low}}^{(\ell)} S_{\text{low}}^{(\ell)} (V_{\text{low}}^{(\ell)})^T$,
>
> where $W_{\text{high}}^{(\ell)}$ is frozen and only $W_{\text{low}}^{(\ell)}$ is trainable.
>
> The raw gradients of the loss $L_t$ w.r.t. the low-rank SVD components are
>
> $G_{U,\text{raw}}^{(\ell)} = \partial L_t / \partial U_{\text{low}}^{(\ell)}$,
>
> $G_{S}^{(\ell)} = \partial L_t / \partial S_{\text{low}}^{(\ell)}$,
>
> $G_{V,\text{raw}}^{(\ell)} = \partial L_t / \partial V_{\text{low}}^{(\ell)}$.
>
> Steps 16–17 project these gradients so that updates lie orthogonal to the frozen high-rank subspaces:
>
> $G_{U}^{(\ell)} = (I - U_{\text{high}}^{(\ell)} (U_{\text{high}}^{(\ell)})^T) G_{U,\text{raw}}^{(\ell)}$,
>
> $G_{V}^{(\ell)} = (I - V_{\text{high}}^{(\ell)} (V_{\text{high}}^{(\ell)})^T) G_{V,\text{raw}}^{(\ell)}$.
>
> The optimizer then applies these updates to the weights. So in our algorithm:
>
> **(1)** The only constraint enforced during training is that updates remain orthogonal to the preserved subspaces:
> $(U_{\text{high}}^{(\ell)})^T G_{U}^{(\ell)} = 0$,
> $(V_{\text{high}}^{(\ell)})^T G_{V}^{(\ell)} = 0$.
>
> **(2)** We do not enforce that $U_{\text{low}}^{(\ell)}$ and $V_{\text{low}}^{(\ell)}$ remain internally orthogonal. They only need to remain orthogonal to $U_{\text{high}}^{(\ell)}$ and $V_{\text{high}}^{(\ell)}$.
>
> This is intentional: we allow the data from the task to decide how best to use the low-rank capacity. If the optimal solution for a task lives in a smaller subspace (i.e., the effective rank of $W_{\text{low}}$ becomes $< r_{\text{low}}$), we do not force it to span a full orthonormal basis. We do not see “full rank $<$ matrix size” as an issue.
>
> At the end of training, we discard the SVD parameterization and reconstruct the standard weight matrix
> $W_{\text{final}}^{(\ell)} = U_{\text{high}}^{(\ell)} S_{\text{high}}^{(\ell)} (V_{\text{high}}^{(\ell)})^T + U_{\text{low}}^{(\ell)} S_{\text{low}}^{(\ell)} (V_{\text{low}}^{(\ell)})^T$,
> and save it in the original model format. All SVD-specific factors and hooks are training-time artifacts only; the final model has the same architecture and parameter shapes as the base model.
>
> ---
>
> **Question: Is $( W \neq W_{\text{high}} + W_{\text{low}} )$?**
>
> In our method we **do** have $W = W_{\text{high}} + W_{\text{low}}.$
>
> We **do not** run two separate SVDs (rank-$k$ and rank-$(n-k)$). We run **one SVD per matrix** and then split the factors according to the importance metric using cosine similarity. So the computation cost is exactly that of a single rank-$n$ SVD.

---

> ### Comment · Reviewer_5uQ5 · 2025-11-27
>
> Thank authors for taking the time to answer my follow-up questions. They addressed most of my concerns, and I would like to raise my score accordingly.

---

### Author Response · Authors · 2025-11-20
**Overall response and revision summary**

We thank all the reviewers for their thoughtful and constructive feedback. We have significantly improved clarity, provided additional experiments and baselines, and tried to address each concern raised. Detailed responses to reviewers are added as comments on OpenReview, outlining the key revisions and clarifications. All new or revised content in the updated manuscript is highlighted in green.

---

### Author Response · Authors · 2025-11-30
**Summary of Discussion Period and Revisions**

Respected Area Chairs,

Overall the review period went very well. The reviews were high quality and very useful in improving the quality of the paper. We have uploaded a revised manuscript where all changes made during the rebuttal are highlighted in green.

Below is a summary of the feedback and how we addressed the specific points raised by each reviewer:

**Reviewer 5uQ5**
* **Feedback:** The reviewer appreciated the adaptive budget allocation but initially raised concerns about computational efficiency (they assumed we calculated full SVD at every step), the motivation for cosine similarity in determining effective rank of a matrix, and missing baselines.
* **Our Response:** We clarified that SVD is computed only once per task (adding negligible overhead) and that training uses a LoRA-style parameterization for efficiency. We added comparisons to **CorDA**, **SVD-LLM**, and **SAPT** (our method performed better than these baselines too) and included task-order robustness results. We also provided ablations for the retention ratio hyperparameters.

**Reviewer oGvj**
* **Feedback:** The reviewer questioned the validity of using top singular vectors as a proxy for high-curvature directions without internal empirical verification. They also raised concerns that our performance gains might stem simply from activating more parameters rather than the method itself and noted the absence of LoRA-based baselines on the TRACE benchmark.
* **Our Response:** We provided new experimental analysis on LLaMA-2 demonstrating a strong correlation between the SVD subspaces, Hessian directions, and Fisher information, validating our curvature proxy. To address the parameter budget concern, we highlighted our ablation study (Appendix A.9) where removing the orthogonality constraint while keeping the parameter count fixed caused accuracy to drop from 79.6% to 31.2%. We also added the requested LoRASeqFT baseline to the TRACE results, where it performed significantly worse than OSFT (9.2% vs 48.4% average accuracy).

**Reviewer uSxf**
* **Feedback:** The reviewer emphasized the strengths and contribution of the paper - "thoughtful and well-motivated", "paper includes computational analysis as well as theoretical justification", and "strong set of comparison methods is included". They suggested the O-LoRA trade-off curve in Figure 1 needed more rigorous tuning (regularization strength) and pointed out missing recent baselines like **MiLoRA** and **PISSA**. They also asked about long-horizon task behavior.
* **Our Response:** We updated Figure 1 to include O-LoRA runs with varying regularization strengths, confirming OSFT still dominates the Pareto frontier. We implemented and compared against MiLoRA and PISSA, outperforming both on math reasoning tasks. We also clarified the "small buffer" limitation regarding data access from the immediate previous task.

**Reviewer zn3T**
* **Feedback:** The reviewer strongly supported the paper, highlighting the "stability-plasticity" mechanism and the theory-practice loop. Their main critique was the lack of evaluation on newer model families (e.g., Qwen) and broader capability checks.
* **Our Response:** We ran new experiments using **Qwen2.5-3B-Instruct** on the TRACE benchmark, where OSFT continued to outperform standard baselines. We also discussed and clarified general capability metrics as suggested by the reviewer.

**Current Status:**
Two reviewers responded with their final scores:
* **Reviewer zn3T** decided to maintain their positive evaluation, stating that most of their concerns have been addressed.
* **Reviewer 5uQ5** stated that we addressed most of their concerns and explicitly confirmed they would raise their score accordingly.

We addressed **Reviewer uSxf’s** comments, including their follow-up questions regarding additional O-LoRA points in the pareto-frontier plot and access to previous task data. While they thanked us for the detailed rebuttal, they did not post a final confirmation after our last response. **Reviewer oGvj** has not yet responded to the rebuttal, but we believe we have fully addressed their concerns regarding the curvature proxy and novelty of our method.

We believe the reviewers received our paper well and were trending towards a positive overall response. We hope this summary helps with the meta-review process.

Best Regards,
The Authors

---

### Meta-Review · Area_Chair_hXqP · 2025-12-21

**Summary:**

The paper introduces Orthogonal Subspace Fine-Tuning (OSFT), a novel PEFT method for continual learning. OSFT uses SVD to separate critical knowledge-bearing directions (high singular values) from underutilized capacity (low singular values). Updates for new tasks are constrained to be strictly orthogonal to the space spanned by the knowledge-bearing directions. This approach effectively balances the plasticity needed for new tasks with stability to retain prior knowledge. The authors also use the similarity between layers' activations to determine the updating ranks of the matrix.

**Reviewer Concerns:**

• The SVD computations may introduce significant overhead.

• Several important baselines are missing, including SAPT, CorDA, InfLoRA, and other LoRA-based approaches.

• Related work—especially MiLoRA—is not sufficiently discussed.

• Experiments on newer model families (e.g., Qwen) are absent.

The authors have addressed some concerns during the rebuttal period.

**Reviewer Scores:**

The paper received ratings of 2, 4, 6, and 8. Reviewer 5uQ5 (score 2) appeared to misunderstand a key detail of OSFT, mistakenly believing that the method updates full-dimensional weights. The authors clarify a misunderstanding the reviewer confirms that the score is raised.

Reviewer oGvj (score 4) primarily questioned whether OSFT’s performance gains might simply result from activating more parameters, and also noted that detailed tuning protocols for baselines were not provided. In response, the authors conducted an ablation study removing the orthogonality constraint and supplied detailed tuning protocols in the revised version. I feel that most of the reviewer's concerns have been addressed.

Overall, I believe that most of the four reviewers' concerns have been adequately addressed.

---

### Decision · Program_Chairs · 2026-01-26

Accept (Poster)